

# The GIEMS-MethaneCentric database: a dynamic and comprehensive global product of methane-emitting aquatic areas

Juliette Bernard[1,2], Catherine Prigent[1,3], Carlos Jimenez[3,1], Etienne Fluet-Chouinard[4], Bernhard Lehner[5], Elodie Salmon[2], Philippe Ciais[2], Zhen Zhang[6], Shushi Peng[7], and Marielle Saunois[2]

[1]LERMA, Paris Observatory, CNRS, PSL, Paris, France
[2]Laboratoire des Sciences du Climat et de l'Environnement, CEA-CNRS-UVSQ, Gif-sur-Yvette, France
[3]Estellus, Paris, France
[4]Earth System Sciences Division, Pacific Northwest National Laboratory, Richland, WA, USA
[5]Department of Geography, McGill University, Montreal, QC H3A 0B9, Canada
[6]Institute of Tibetan Plateau Research, Chinese Academy of Sciences, Beijing, China
[7]College of Urban and Environmental Sciences, Peking University, Beijing 100871, China

**Correspondence:** Juliette Bernard (juliette.bernard@obspm.fr) and Catherine Prigent (catherine.prigent@obspm.fr)

**Abstract.**

The Global Inundation Extent from Multi-Satellites (GIEMS) database first published in 2001 (Prigent et al., 2001) was a key advance toward the accurate representation of wetlands globally by providing dynamic time series of global surface water based on passive microwave observations. This study supplements the second version of GIEMS (GIEMS-2) with other

datasets to produce GIEMS-MethaneCentric (GIEMS-MC), a dynamically mapped dataset of methane-emitting waterlogged and inundated ecosystems. We separated open water from wetlands in GIEMS-MC by using the Global Lakes and Wetlands Database version 2 (GLWDv2), while adding unsaturated peatland areas undetected by GIEMS-2. Rice paddies are identified using the Monthly Irrigated and Rainfed Crop Areas (MIRCA2000) product. A specific coastal zone filtering is applied to avoid ocean artifacts while preserving coastal wetlands. GIEMS-MC covers the period 1992-2020 on a monthly scale at 0.25°x0.25°

spatial resolution. The GIEMS-MC product includes two layers of monthly wetland time series - one for flooded and saturated wetlands and another for all wetlands and peatlands - together with seven layers of compatible static maps of open water bodies (lakes, rivers, reservoirs) and seasonal rice paddy maps used in its production. The dominant vegetation and wetland types per pixel are also provided in GIEMS-MC variables. GIEMS-MC is compared to Wetland Area and Dynamics for Methane Modelling (WAD2M), a dataset providing dynamic wetland information. In terms of wetland extent, GIEMS-MC

all wetlands and peatlands and WAD2M show similar results, with a mean annual maximum of 7.8 Mkm$^2$ for GIEMS-MC and 6.8 Mkm$^2$ for WAD2M, and similar spatial patterns in most regions. The GIEMS-MC seamless time series represents a significant advance in wetland representation for methane modelling, although limitations remain in the accurate identification of rice, coastal and peatland areas. This resource provides harmonized dynamic maps of aquatic methane emitting surfaces and is available at https://zenodo.org/records/13919645.





## 1  Introduction

Following a stable period from 1999 to 2006, atmospheric methane levels have started to rise again, reaching a record growth rate of +18 ppb yr$^{-1}$ in 2021 (Lan et al., 2024). This increase is a cause for concern, particularly given that anthropogenic emissions of this potent greenhouse gas account for approximately one-third of the human-induced radiative forcing (Szopa et al., 2021). As a chemically active greenhouse gas with multiple, time-varying sources and sinks (Saunois et al., 2024),

closing the methane budget is challenging. The causes of the observed increase in atmospheric methane remain uncertain. Potential factors include increased human or natural emissions, reduced sinks, or a combination of these factors. However, isotopic evidence suggests that biogenic sources (livestock, wetlands, waste, etc.) may play a significant role in the observed increase (Nisbet et al., 2016, 2019).

Among the sources, natural emissions from wetlands and freshwater ecosystems account for 145 to 369 Tg CH$_4$ yr$^{-1}$, i.e.,

25 to 51% of global methane emissions (Saunois et al., 2024). Wetland emissions show significant inter-annual variability (Bousquet et al., 2006; Bridgham et al., 2013) and are sensitive to climate (Bridgham et al., 2013; Zhang et al., 2023). Thus, better understanding natural methane emissions variability in the past will inform future predictions of wetland emissions and their feedback on climate. Large uncertainties remain for both wetlands and freshwater ecosystems methane emissions (Saunois et al., 2020; Canadell et al., 2021). This is due to the difficulty of modelling methane fluxes, which depend on many biotic and

abiotic factors (Bridgham et al., 2013; Ge et al., 2024), to the small number of flux observations (Canadell et al., 2021), and to uncertainties in wetland and freshwater area (Bridgham et al., 2013; Melton et al., 2013; Saunois et al., 2020; Canadell et al., 2021), including issues of double counting, where the same area may be counted twice under different categories, inflating estimated emissions (Canadell et al., 2021; Thornton et al., 2016). Yet, the area covered by seasonal wetlands remains the single largest source of uncertainty on wetland CH$_4$ emissions (Melton et al., 2013; Peltola et al., 2019; Poulter et al., 2017;

Zhang et al., 2017).

The first global wetland map was produced by Matthews and Fung (1987), providing composite static information on wetland types. Since then, new static wetland products have been established, either from composite information (Lehner and Döll, 2004; Tootchi et al., 2019; Tuanmu and Jetz, 2014) or from remote sensing approaches (Loveland et al., 2000; Friedl et al., 2002; Bartholomé and Belward, 2005; Carroll et al., 2009; Feng et al., 2016). Further datasets have been developed based

on hydrological model outputs (Ringeval et al., 2012; Wania et al., 2013; Xi et al., 2022), presenting their advantages and disadvantages compared to satellite-derived products. Those models can be used both to reconstruct the historical distribution of wetlands and to predict their future evolution. Modelling can be an effective method for producing a global map of wetlands, particularly where physics-based models can reflect the mechanisms by which wetlands are formed. The two main limitations of these model outputs are 1) that hydrological models are simplified representations of the real-world complexity of wetlands

(e.g., models often focus on a single water surface generation process (Obled and Zin, 2004)), and 2) that human interference is not well accounted for in the models (Hu et al., 2017). Moreover, observations are required to constrain and/or validate these model predictions.





However, there are only a few available observational dynamic time series of surface water maps at a global scale. Notably, these include: 1. the Global Inundation Extent from Multi-Satellites (GIEMS and GIEMS-2) (Prigent et al., 2001, 2007; Papa et al., 2010; Prigent et al., 2020) and its downscaled versions (Fluet-Chouinard et al., 2015; Aires et al., 2017), and 2. the Surface Water Microwave Product Series (SWAMPS) (Schroeder et al., 2015; Jensen and Mcdonald, 2019).

GIEMS-2 and SWAMPS both provide monthly fractions of surface water at 0.25°x0.25° for 1992-2020, mainly based on passive microwave observations from Special Sensor Microwave Imager (SSM/I) and the Special Sensor Microwave Imager Sounder (SSMIS). Although SWAMPS and GIEMS-2 both aim to represent both inundated surfaces and are produced using similar input data, they present significant differences both in terms of spatial distribution and inter-annual variations (Pham-Duc et al., 2017; Bernard et al., 2024b).

GIEMS-2 and SWAMPS products do not differentiate surface water categories, e.g., wetland, lake, reservoir, pond, or rice paddy, and are therefore not directly usable for wetland studies modeling seasonally inundated wetlands separately from open water bodies. Recent efforts have been made by Zhang et al. (2021b) to produce the Wetland Area and Dynamics for Methane Modeling (WAD2M) product based on SWAMPS, which represents a pioneering attempt to dynamically map wetlands, including peatlands. Using additional high-resolution static estimates of wetlands and open permanent water, as well as seasonal information on rice paddies, Zhang et al. (2021b) were able to apply these correction layers to SWAMPS to distinguish wetlands from other surface water. WAD2M version 2.0 (Zhang et al., 2021a) provides monthly estimates on a global scale for 2000-2020 at 0.25°x0.25°. However, WAD2M has encountered difficulties in capturing reliable inter-annual trends (Zhang et al., 2021b; Bernard et al., 2024b). In fact, issues in SWAMPS are propagated into WAD2M, such as ocean/desert artifacts leading to overestimation and abrupt changes in time series partly due to changes in satellites (Pham-Duc et al., 2017; Bernard et al., 2024b).

In an attempt to compare the corrected wetland extent of WAD2M and GIEMS, McNicol et al. (2023) applied the same correction layers to GIEMS-2, but this exercise did not eliminate the large differences between these two datasets. In particular, the WAD2M procedure rescales the SWAMPS surface water extent fractions, which are always positive, with other high resolution static wetland datasets, which potentially produces some unreliable seasonality where the wetland fractions in SWAMPS are below the instrumental noise level. On the contrary, GIEMS-2 shows some zero fractions over numerous pixels where no water is detected. This makes it impossible to use the same procedure as in Zhang et al. (2021b) to produce WAD2M from SWAMPS. The correction procedure needs to be modified to adapt to GIEMS-2. Furthermore, since the release of WAD2M, the most recent maps of aquatic ecosystems have been aggregated into the Global Lakes and Wetlands Database version 2 (GLWDv2), which now offers the most comprehensive and up-to-date representation of global wetland classes (Lehner et al., 2024a).

This study presents a new comprehensive database of methane emitting surfaces, named the Global Inundation Extent from Multi-Satellites-MethaneCentric (GIEMS-MC). GIEMS-MC aims at providing spatially and dynamically consistent maps of the different methane-emitting ecosystems, with the purpose of providing data for modelling methane emissions at the global scale (0.25°x0.25°) over 1992-2020. In particular, two time series of wetland maps are developed at monthly timescale: inundated and saturated wetlands (ISW), and all wetlands including non-inundated peatlands (inundated and saturated wetlands





+ peatlands, ISW+P). GIEMS-MC also provides compatible information from the ancillary data used, including static surface

extents of open permanent waters (lakes, rivers, reservoirs), and seasonal surface extents of rice paddies, along with dominant

vegetation on wetland classes. GIEMS-MC takes advantage of the GIEMS-2 product that offers ∼30-year seamless time

series of surface water with realistic seasonality and inter-annuality (Prigent et al., 2020; Bernard et al., 2024b), and largely

benefits from the recently developed comprehensive static map of GLWDv2 (Lehner et al., 2024a). This article outlines the

methodology behind the production of GIEMS-MC and provides an analysis and comparison with existing datasets : WAD2M

(Zhang et al., 2021b), GLWDv2 (Lehner et al., 2024a), and the original GIEMS-2 (Prigent et al., 2020). Two inundation

products based on Cyclone Global Navigation Satellite System (CYGNSS) data are also used for comparison over the Sudd

(Zeiger et al., 2023; Gerlein-Safdi et al., 2021). The sensitivity of the wetland estimates to the different process steps is also

discussed.

## 2  Datasets

This section presents the three types of data used in the production of GIEMS-MC: 1. the surface data of the different aquatic

ecosystems, 2. the data used for the masks and additional ecosystem layers, and 3. the WAD2M comparison dataset.

### 2.1  Input datasets to GIEMS-MC (GIEMS-2, GLWDv2, MIRCA2000)

The GIEMS-2 dataset, spanning 1992 to 2015 and extended to 2020 in this study, uses mainly passive microwave observations

from the SSM/I and SSMIS satellites at frequencies from 19 to 85 GHz, as described in Prigent et al. (2020). This dataset

utilizes also active microwave satellite data and Normalised Difference Vegetation Index (NDVI) derived from visible and

near-infrared measurements to characterize vegetation and mitigate its influence on the passive microwave signal. The initial

GIEMS-1 methodology (Prigent et al., 2001, 2007) has been thoroughly evaluated (Papa et al., 2006; Prigent et al., 2007; Papa

et al., 2008, 2010), as was the new GIEMS-2 algorithm (Prigent et al., 2020; Bernard et al., 2024b). GIEMS-2 provides monthly

global maps of surface water extent with a spatial resolution of 0.25°x0.25°. The continuity of this dataset relies on carefully

intercalibrated SSM/I and SSMIS observations (Fennig et al., 2020). GIEMS-2 includes all continental water surfaces, such

as wetlands, rice paddies, rivers, reservoirs, and lakes, with the exception of large lakes (> 15 000 km$^2$), which have been

masked out. Microwave observations used in GIEMS-2 are sensitive to the presence of snow, and this contamination prevents

the calculation of surface water over snow-covered regions. Thus, snow-covered pixels are set to 0 fraction using ERA5 in the

previous studies and in the distributed GIEMS-2 product. Passive microwaves are sensitive to the presence of water, including

estuarine and offshore marine waters. To avoid misinterpretations of the data, coastal pixels have been filtered out from the

distributed GIEMS-2 product, leading to possible underestimation of inundated surface extent in the coastal areas. Here we

use an unfiltered version of GIEMS-2 in which coastal regions are not excluded, in order to improve the cleaning of the coasts

during the production process of GIEMS-MC based on GLWDv2, as described in Sect. 3.

The Global Lakes and Wetlands Database version 2 (GLWDv2) (Lehner et al., 2024a) provides comprehensive global maps

of aquatic ecosystems synthesized from a variety of ground- and satellite-based data products. GLWDv2 combines various data



products to generate consolidated and harmonized static maps representative of the period 1990-2020. The GLWDv2 product
contains 33 wetland and water body classes, which are listed in Supplementary Table S1. GLWDv2 represents the maximum
extent of each of its 33 classes (in pixel fraction) at a resolution of 15 arc seconds (approximately 500 m at the equator). For
this study, the 33 GLWDv2 class maps were aggregated at 0.25°x0.25°.

Rice cultivation varies seasonally according to cropping calendars, and their inundated cover can be confused with that of
wetlands. The majority of global rice paddy maps are static representations, typically for a specific time period. A notable
data source that gives insights into the seasonality of rice paddy at global scale is the MIRCA2000 dataset (Portmann et al.,
2010). MIRCA2000 provides data on irrigated and rainfed cultivated areas at a resolution of 5 arc minutes for each month
of a reference year (representative of circa 2000). The dataset integrates several data sources, including agricultural statistics
such as cropping calendar and remote sensing data. This study uses both irrigated and rainfed rice data extracted from the
MIRCA2000 dataset.

## 2.2   Ancillary and correction datasets (ERA5, ESA CCI)

The European Centre for Medium-range Weather Forecasts reanalysis (ECMWF-ERA5) (Hersbach et al., 2020) is a state-
of-the-art reanalysis for climate applications. It provides global climate and weather data spanning from 1940 to the present.
ERA-5 uses assimilation techniques by integrating a wide diversity of observational data to deliver hourly estimates of multiple
atmospheric, land, and oceanic variables at a resolution of 31 km. ERA5 can be downloaded at a resolution of 0.25° x 0.25°
from https://cds.climate.copernicus.eu/. In the GIEMS-2 and GIEMS-MC process, the area covered with snow in a pixel is
derived from the ERA5 variables snow density and snow depth.

The European Space Agency (ESA) Climate Change Initiative (CCI) Land Cover dataset (ESA, 2017) provides a classifica-
tion of land cover features at a spatial resolution of 300 m for each year from 1992 to 2022. The dataset is derived from various
satellite Earth observation data. According to the standards of the United Nations Land Cover Classification System (Di Gre-
gorio and Jansen, 2005), it contains 22 land cover classes (Supplementary Table S2), including 18 vegetation categories and
urban, bare, water bodies, and snow/ice categories. The ESA CCI Land Cover dataset can be accessed via the ESA CCI Land
Cover project website : https://maps.elie.ucl.ac.be/CCI/viewer/download.php. Here, we aggregated a version to 0.25° x 0.25°,
where the dominant class within each pixel is determined based on the highest fractional coverage.

## 145   2.3   Comparison dataset (WAD2M)

The Wetland Area and Dynamics for Methane Modeling (WAD2M) version 2.0 dataset (Zhang et al., 2021a, b) is a compre-
hensive global product designed to support methane modelling. It provides the fraction of wetland area, including peatlands, at
a resolution of 0.25°x0.25°, and at a monthly time step for 2000-2020. The WAD2M dataset uses dynamic data from the Sur-
face Water Microwave Product Series (SWAMPS) dataset (Jensen and Mcdonald, 2019), which provides monthly inundation
fraction at 0.25°x0.25°. Similar to GIEMS-2, SWAMPS is derived mainly from passive microwave observations from SSM/I
and SSMIS, but the methodology and ancillary data used differ between the two products (Schroeder et al., 2015; Prigent
et al., 2020), resulting in important differences in some regions (Pham-Duc et al., 2017; Bernard et al., 2024b). The creation

Earth System
Science
Data

of WAD2M involved combining SWAMPS surface inundation time series with static datasets to distinguish between different wetland types. The static datasets used in WAD2M production are 4 peatlands maps (NCSCD from Hugelius et al. (2013),

CAWASAR from Widhalm et al. (2015), GLWDv1 from Lehner and Döll (2004), and CIFOR from Gumbricht et al. (2017)), one inland open water map (GSW from Pekel et al. (2016)), one coastal mask (MOD44W from Carroll et al. (2009)), and seasonal irrigated rice map (MIRCA2000 from Portmann et al. (2010)). These static layers allow the wetland fractions to be rescaled to include non-inundated wetlands (peatlands) and exclude non-wetland inundated areas (irrigated rice paddies and open waters).

# 3    Methods

## 3.1    Overview of the methodology

GIEMS-2 uses satellite passive microwave data, which are particularly responsive to the presence of water, to determine the fraction of inundated and saturated soil per pixel. However, modifications to the GIEMS-2 dataset are required in order to remove inundated or saturated areas that are not wetlands (e.g., rice paddies, lakes, rivers, reservoirs), and to add wetlands

where the water table may be undetectable below ground level (e.g., some peatlands). As a consequence of the aforementioned remote sensing approach, the present study will first distinguish the inundated wetlands identified by GIEMS-2 and then add the unsaturated wetlands. In addition to GIEMS-2, the GLWDv2 dataset will be used.

The original GIEMS-2 product (Prigent et al., 2020) has been extended to 2020 (Bernard et al., 2024b), and a special version without coastal filtering is used here. In total, seven steps, described in the following subsections, are required to derive wetland

maps from this data. The operations are made in terms of pixel fraction $f$ on a regular grid of 0.25°x0.25°. Multiplication by pixel area is then needed to derive wetland extent. A summary of the procedure is shown in Fig. 1, and the seven steps are described in detail in the following subsections.

### 3.1.1    Applying ocean mask

For consistency with GLWDv2, we here used the regional shapefiles of the HydroATLAS database (version 1.0; Linke et al.

(2019)), which provides near-identical coastlines as GLWDv2. This allowed us to calculate the ocean fraction for each 0.25° x 0.25° pixel. The ocean water fraction is set to -999 if the ocean fraction of a pixel is greater than 99%, to avoid confusion between ocean pixels and pixels where no surface water was detected (zero fraction pixels).

### 3.1.2    Applying snow mask

GIEMS-2 surface water detection relies primarily on passive microwave observations, which are affected by the presence of

snow (Foster et al., 1984). Thus, the surface water fraction cannot be reliably quantified in the presence of snow. Consequently, the surface water detection algorithm in the GIEMS-2 production is not run when snow is present in a pixel. To exclude these snow-covered pixels, ECMWF snow information from ERA5 is used in the GIEMS-2 processing, and pixels with a snow



**Figure 1.** Schematic of the GIEMS-MC dataset production process. All operations are performed in terms of pixel fractions at a resolution of 0.25°x0.25°. Ocean pixels are set to -999, snow-covered pixels are set to -998, and urban pixels are set to -997 in a revised version of GIEMS-2. Open permanent water and then rice paddies areas are subtracted from the surface water areas. A specific coastal cleaning is applied to remove ocean contamination, resulting in a dynamic map of Inundated and Saturated Wetland (GIEMS-MC$_{ISW}$). Peatlands areas undetected by GIEMS-2 are added to derive a dynamic map of all wetlands including peatlands, called Inundated and Saturated Wetland + Peatland (GIEMS-MC$_{ISW+P}$). Finally, initial ancillary data information is added to the product so that users can easily access the different fraction maps of all surface water categories, including Inundated and Saturated Wetland, Inundated and Saturated Wetland + Peatland, open permanent waters, rice paddies, and the dominant wetland and vegetation classes. $f_i$ refers to the fraction of a pixel $i$ before the corresponding step. pos($f$) refers to the positive part of $f$, i.e. pos($f$) = max($f$,0). GLWDv2 Open Permanent Water (GLWDv2$_{OPW}$) is the sum of all GLWDv2 classes 1 to 5. GLWDv2 Inundated and Saturated Wetland (GLWDv2$_{ISW}$) is the sum of all GLWDv2 wetlands excluding peatlands corresponding to classes 8 to 21, 28, 29, 31, and 32. GLWDv2 Peatland (GLWDv2$_{peat}$) is the sum of all GLWDv2 peatlands corresponding to classes 22 to 27.

fraction above 2% are set to a surface water fraction of 0 (Prigent et al., 2020). In GIEMS-MC, the pixel value is given its dedicated snow flag value of -998 when the snow fraction of a pixel is greater than 2%. It should be noted that this mask

remains for all subsequent steps and is therefore also applied to the peatlands (step 7).

### 3.1.3 Applying urban mask

It has been observed that unexpectedly large water surfaces are detected by GIEMS-2 in areas of high urban density. This could be due to the different surface materials used in buildings, some of which strongly reflect microwaves. For example, highly reflective areas over Paris are misinterpreted as water due to predominance of zinc roofs. To apply an urban mask, the urban

class product of the ESA CCI land cover map aggregated at 0.25°x0.25° is used. The grid cells with urban percentage above 40% are systematically masked to -997 to avoid any confusion between urban and water surfaces. Note that applying this urban mask results in neglecting change in terms of surface water global area (<1% change on mean extent), but this avoids local artifacts over the high urban density areas.

### 3.1.4 Subtracting open permanent waters

Inland permanent open waters are considered separately from wetlands in methane budgets (Saunois et al., 2020; Canadell et al., 2021), as different methane production and transport processes are involved. To derive wetland maps, these open permanent surface water areas must be subtracted from the GIEMS-2 estimates. Here, we define permanent open water as non-vegetated, permanently inundated areas that are not wetland. Some dynamic datasets could have been used, but consistency was preferred, so GLWDv2 harmonized maps were used in GIEMS-MC production. Then, permanent open water areas of GLWDv2 corre-

sponding to layers 1 to 5 (*Freshwater Lake*, *Salt Lake*, *Reservoir*, *Large River* and *Large Estuarine River*) are subtracted from the GIEMS-2 fractions. These GLWDv2 areas are derived from HydroLAKES (Messager et al. (2016) ; Lakes), the Global Dam Watch (GDWv1) database (Lehner et al. (2024b) ; Reservoirs), the Global River Width from Landsat (GRWL) dataset (Allen and Pavelsky (2018) ; Large Rivers) and augmented with the Global Surface Water (GSW) database (Pekel et al., 2016)).

### 3.1.5 Subtracting rice paddies

Rice paddies are intermittently saturated or inundated depending on irrigation practices, and their methane emissions are considered to be an anthropogenic source that should be separated from those of natural wetlands. GLWDv2 contains a static rice paddy map, but the seasonal variation of rice paddies is important in terms of extent and needs to be taken into account to avoid over-subtraction of rice paddies in the GIEMS-MC process. However, there is to our knowledge no dynamic (intra-annual resolution) product available that represents rice paddies at global scale over our observation period. As the MIRCA2000

product provides maps with a typical seasonality (circa 2000) of global rice paddies, it appears to be the most appropriate product available. Consequently, the MIRCA2000 12-month seasonality of irrigated and rainfed rice paddy areas is subtracted from the area estimates. This rice paddy processing and its uncertainties are discussed further in Sect. 5.2.2.



### 3.1.6 Correcting ocean contamination

The GIEMS-2 version used here has not been filtered in coastal areas, as it is usually done in the distributed GIEMS-2 version.
The SSM/I and SSMIS passive microwave observations used in GIEMS-2 production are very sensitive to the presence of water, including the ocean. The GIEMS-2 fraction and seasonality estimates are less reliable for pixels with larger ocean fractions. Thus, pixels containing more than 10% ocean in GLWDv2 (GLWDv2$_{ocean}$> 10%) are set to 0. Tests were made to tune this 10% threshold, to avoid masking all pixels containing a small fraction of ocean area, while ensuring reasonable seasonality. However, pixels containing up to 10% ocean area will undergo an additional coastal cleaning procedure that follows. Ocean
contamination can arise from the presence of ocean within the pixel, but also from the ocean in neighboring pixels. A GIEMS-2 pixel ($\sim$800 km$^2$ at the equator, $\sim$400 km$^2$ at 60°N or S) is smaller than the -3dB footprint of the original microwave satellite observations (69 km$\times$43 km at 19 GHz and 37 km$\times$28 km at 37 GHz). Moreover, microwave energy is also measured in the side lobes of the satellite instrument footprint. Coastal areas should then undergo a cleaning process to reduce these artifacts. Pixels whose centers are between 0 and 50 km from a coastline or large lakes (> 15 000 km$^2$) are considered as coastal areas.

In an attempt to correct for ocean contamination, the following procedure is applied to ensure that the wetland areas in GIEMS-MC in coastal regions are equal to or lower than the GLWDv2 inventory. This is done by calculating GLWDv2 Inundated and Saturated Wetland (GLWDv2$_{ISW}$), i.e., the sum of all GLWDv2 wetlands excluding peatlands (which are not necessarily saturated surface water) corresponding to classes 8 to 21, 28, 29, 31, and 32. The fraction of the modified version of GIEMS-2 up to this step, incorporating the 5 aforementioned corrections and cleaning processes (ocean, snow, urban masks and
open water, and rice paddies removal), is called $f$. Its Mean Annual Maximum $f_{MAmax}$ is calculated by taking a monthly average over all years and taking the maximum of this monthly seasonality for each pixel $i$. In the coastal region, for each time step and for each pixel $i$, with the resulting areas called Inundated and Saturated Wetland map in GIEMS-MC (GIEMS-MC$_{ISW}$):

- if GLWDv2$_{ocean, i}$ > 10%:
  then GIEMS-MC$_{ISW, i}$ = 0

- if GLWDv2$_{ocean, i}$ $\leq$ 10% and $f_{MAmax, i}$ < GLWDv2$_{ISW, i}$:
  then GIEMS-MC$_{ISW, i}$ = $f_i$

- if GLWDv2$_{ocean, i}$ $\leq$ 10% and $f_{MAmax, i}$ > GLWDv2$_{ISW, i}$:
  then GIEMS-MC$_{ISW, i}$ = $f_i * \frac{\text{GLWDv2}_{ISW, i}}{f_{MAmax, i}}$

### 3.1.7 Adding peatlands

Finally, in order to have a complete map of wetlands, the peatlands not detected by GIEMS-2 (monthly unsaturated or unflooded peatlands) have to be taken into account in the wetland fraction. This is done using the following procedure. The sum of GLWDv2 peatlands, i.e. GLWDv2 classes 22 to 27, is denoted here as GLWDv2$_{peat}$. GLWDv2 peatland information is a composite product relying on most up-to-date peatland maps : PeatMap (Xu et al. (2018), global), SoilGrids250m (Hengl et al. (2017), global), Northern Peatlands (Hugelius and Olefeldt, north of 23° N), and CIFOR (Gumbricht et al. (2017), only





south of 23.5° N). More details can be found in Lehner et al. (2024a). GLWDv2$_{peat}$ represents 4.26 Mkm$^2$, which is consistent with primary PeatMap estimates of 4.23 Mkm$^2$. The peatlands detected by GIEMS-2 are derived by the difference, if positive, between GIEMS-MC$_{ISW}$ and GLWDv2$_{ISW}$. The undetected peatlands are then derived for each month as the difference between GLWDv2$_{peat}$ and the peatlands detected by GIEMS-2. These undetected peatlands are added to the GIEMS-MC$_{ISW}$, resulting in GIEMS-MC Inundated and Saturated Wetland + Peatland (GIEMS-MC$_{ISW+P}$), i.e., for each pixel $i$ :

$$\text{undetected peatlands}_i = \text{pos}[\text{GLWDv2}_{peat\,i} - \text{pos}(\text{GIEMS-MC}_{ISW\,i} - \text{GLWDv2}_{ISW\,i})\,]$$
$$\text{GIEMS-MC}_{ISW+P} = \text{GIEMS-MC}_{ISW} + \text{undetected peatlands.}$$

The uncertainty in terms of areas of this step is discussed in Sect. 5.2.3.

## 3.2 Comparison

GIEMS-MC$_{ISW}$ is compared with the original GIEMS-2 product and GLWD$_{ISW}$. GIEMS-MC$_{ISW+P}$ is compared to GLWDv2$_{ISW+P}$
(GLWDv2$_{ISW}$ + GLWDv2$_{peat}$) and to WAD2M, as all also include peatlands (Table 3, Fig.2, and Supplementary Fig.S1). As GLWDv2 is a static map representing long term maximum extent, the GIEMS-MC Long Term maximum (LTmax) will be used for comparison with GLWDv2 instead of MAmax (Table 3). To derive this LTmax, the maximum of each pixel over the whole time period is selected. This can lead to the selection of extreme values with moderate reliability, and LTmax should then be interpreted with caution.

## 3.3 Description of GIEMS-MC dataset

Following the seven steps outlined, a netcdf product at 0.25°x0.25° resolution containing the derived variables and ancillary variables is created. The variables included in this product are detailed in Table 1. Its components include both dynamic monthly maps of GIEMS-MC$_{ISW}$ and GIEMS-MC$_{ISW+P}$. Permanent Open Water classes (GLWDv2), i.e., Freshwater Lake, Saline Lake, Reservoir, River, and Estuarine River, are also added as static variables in GIEMS-MC. The 12-month seasonality
of Irrigated and Rainfed Rice Paddy (MIRCA2000) is included. Three static maps provide information on the main ecosystems per pixel : the dominant aquatic class (GLWDv2), the dominant wetland or peatland class (GLWDv2), and the dominant land cover class (ESA CCI Land Cover map).



| GIEMS2-MC variable | Long name | Type | Primary or main data source | Time resolution |
|---|---|---|---|---|
| `inund_sat_wetland_frac` | Inundated and Saturated Wetland | fraction | GIEMS-2 | monthly |
| `inund_sat_peat_wetland_frac` | Inundated and Saturated Wetland + Peatland | fraction | GIEMS-2 + GLWDv2 | monthly |
| `fresh_lake_frac` | Freshwater Lake | fraction | GLWDv2 | static |
| `saline_lake_frac` | Saline Lake | fraction | GLWDv2 | static |
| `reservoir_frac` | Reservoir | fraction | GLWDv2 | static |
| `river_frac` | Large River | fraction | GLWDv2 | static |
| `estu_river_frac` | Large Estuarine River | fraction | GLWDv2 | static |
| `rice_irri_frac` | Irrigated Rice Paddy | fraction | MIRCA2000 | 12-month seasonality |
| `rice_rainfed_frac` | Rainfed Rice Paddy | fraction | MIRCA2000 | 12-month seasonality |
| `dom_aqua_class` | Dominant Aquatic Class | 33 classes (Table S1) | GLWDv2 | static |
| `dom_wet_peat_class` | Dominant Wetland or Peatland Class | 25 classes (Table S1) | GLWDv2 | static |
| `dom_land_cover_class` | Dominant Land Cover Class | 37 classes (Table S2) | ESA CCI Land Cover | static |

**Table 1.** Summary of GIEMS-MC variables with corresponding data sources and temporal resolution. For details about data sources, see Sect. 2 or Prigent et al. (2020) for GIEMS-2, Lehner et al. (2024a) for GLWDv2, Portmann et al. (2010) for MIRCA2000, and ESA (2017) for ESA CCI Land Cover.

## 4 GIEMS-MC results

### 4.1 Global inland water areas

To quantify the variations in terms of extent, the Mean Annual maximum (MAmax), Mean Annual mean (MAmean), and Mean Annual minimum (MAmin) are calculated by averaging the 29-year data to a typical 12-month seasonality for each pixel. Then, the maximum, the mean, and the minimum are respectively selected for each pixel.

   Globally, GIEMS-MC$_{ISW}$ represent 3.90 Mkm$^2$, with a MAmean of 1.27 Mkm$^2$ (Table 2). The addition of peatlands greatly increases these global areas, with GIEMS-MC$_{ISW+P}$ reaching 7.83 Mkm$^2$ (+3.93 Mkm$^2$) in terms of MAmax and 3.54 Mkm$^2$





(+2.27 Mkm$^2$) in terms of MAmean. This increase is mainly due to Europe+Siberia, and North America, where large peatlands contribute significantly to the total wetland area (74% and 58% respectively).

| | | Global | Africa | Asia | Europe +Siberia | Oceania | North America | South America |
|---|---|---|---|---|---|---|---|---|
| **GIEMS-MC$_{ISW}$ (Inundated and Saturated Wetland)** | MAmax | 3895 | 518 | 1034 | 618 | 207 | 837 | 676 |
| | MAmean | 1274 | 166 | 276 | 155 | 103 | 262 | 308 |
| | MAmin | 264 | 48 | 31 | 12 | 46 | 32 | 93 |
| **GIEMS-MC$_{ISW+P}$ (Inundated and Saturated Wetland + Peatland)** | MAmax | 7834 | 638 | 1212 | 2370 | 522 | 1993 | 1019 |
| | MAmean | 3538 | 296 | 403 | 889 | 429 | 794 | 666 |
| | MAmin | 1318 | 183 | 97 | 79 | 378 | 78 | 455 |
| Freshwater lake | static | 2045 | 197 | 76 | 429 | 22 | 1214 | 81 |
| Saline lake | static | 359 | 33 | 97 | 88 | 40 | 21 | 20 |
| Reservoir | static | 316 | 40 | 62 | 71 | 6 | 87 | 47 |
| River | static | 384 | 40 | 72 | 109 | 13 | 53 | 93 |
| Estuarine river | static | 79 | 5 | 12 | 16 | 9 | 10 | 13 |
| Irrigated Rice Paddy | MAmax | 639 | 14 | 509 | 16 | 46 | 14 | 22 |
| | MAmean | 431 | 8 | 355 | 6 | 30 | 8 | 10 |
| | MAmin | 64 | 0 | 62 | 0 | 0 | 0 | 0 |
| Rainfed Rice Paddy | MAmax | 614 | 42 | 452 | 0 | 71 | 3 | 25 |
| | MAmean | 251 | 21 | 178 | 0 | 29 | 1 | 12 |
| | MAmin | 1 | 0 | 0 | 0 | 0 | 0 | 0 |
| Total | MAmax | 12271 | 1013 | 2495 | 3102 | 731 | 3400 | 1326 |
| Total distribution | MAmax | 100% | 8.3% | 20.3% | 25.3% | 6.0% | 27.7% | 10.8% |

**Table 2.** Global and continental surfaces of GIEMS-MC variables in 10$^3$ km$^2$. For dynamic classes, MAmax and MAmin are shown. Total MAmax is the sum of GIEMS-MC Inundated and Saturated Wetland + Peatland MAmax, open permanent water (Freshwater Lake, Saline Lake, Reservoir, River, Estuarine River) from GLWDv2, and Rice Paddy from MIRCA2000 MAmax (Irrigated and Rainfed). Regions correspond to the shapefiles of HydroATLAS database (version 1.0; Linke et al. (2019)).

GIEMS-MC$_{ISW}$ consistently shows much lower extent than the original GIEMS-2 (MAmax reduced from 6.80 Mkm$^2$ to 3.90 Mkm$^2$) that comprises all inundated and saturated areas, including non-wetland categories. The lower areas in GIEMS-MC$_{ISW}$ is mainly due to the removal of open permanent waters in Europe, Siberia, and North America, and to rice paddies

subtraction in Asia. The LTmax of GIEMS-MC$_{ISW}$ reaches 8.90 Mkm$^2$, close to the GLWDv2$_{ISW}$ estimates of 8.22 Mkm$^2$.

Globally, the MAmean estimates of GIEMS-MC$_{ISW+P}$ and WAD2M are in agreement (resp. MAmean of 3.54 Mkm$^2$ and 4.21 Mkm$^2$), but regional differences exist in Africa (MAmean of 296 and 719 10$^3$km$^2$, respectively) and Oceania (MAmean of 429 and 572 10$^3$km$^2$, respectively). In those regions, WAD2M detects comparatively more water, likely due to desert con-

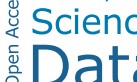



tamination in the SWAMPS product used in the WAD2M production (see Fig.2 and Supplementary Fig.S1). In Asia, Europe,
Siberia, and North America, GIEMS-MC$_{ISW+P}$ shows similar MAmean areas but has a larger MAmax-MAmin amplitude,
possibly due to 1) higher peatland estimates in GLWDv2 than in the ancillary data used in WAD2M production, which could
explain the higher MAmax, and 2) more stringent snow and coastal filtering in GIEMS-MC, which could explain the lower
MAmin. As GLWDv2 peatlands are used to derive GIEMS-MC$_{ISW+P}$ from GIEMS-MC$_{ISW}$, similar total extents are consis-
tently found between GIEMS-MC$_{ISW+P}$ LTmax of 12.24 Mkm$^2$ and GLWDv2$_{ISW+P}$ LTmax of 12.49 Mkm$^2$.

|  |  | Global | Africa | Asia | Europe +Siberia | Oceania | North America | South America |
|---|---|---|---|---|---|---|---|---|
| **GIEMS-MC$_{ISW}$** (Inundated and Saturated Wetland) | LTmax | 8894 | 1414 | 2177 | 1551 | 648 | 1606 | 1493 |
|  | MAmax | 3895 | 518 | 1034 | 618 | 207 | 837 | 676 |
|  | MAmean | 1274 | 166 | 276 | 155 | 103 | 262 | 308 |
|  | MAmin | 264 | 48 | 31 | 12 | 46 | 32 | 93 |
| Orginal GIEMS-2 | MAmax | 6796 | 631 | 1793 | 1071 | 339 | 1804 | 945 |
|  | MAmean | 2730 | 236 | 659 | 339 | 216 | 647 | 506 |
|  | MAmin | 795 | 88 | 149 | 42 | 133 | 93 | 218 |
| GLWDv2$_{ISW}$ (Inundated and Saturated Wetland) | static | 8223 | 1010 | 1958 | 1416 | 541 | 1755 | 1269 |
| **GIEMS-MC$_{ISW+P}$** (Inundated and Saturated Wetland + Peatland) | LTmax | 12374 | 1516 | 2321 | 3116 | 915 | 2639 | 1786 |
|  | MAmax | 7834 | 638 | 1212 | 2370 | 522 | 1993 | 1019 |
|  | MAmean | 3538 | 296 | 403 | 889 | 429 | 794 | 666 |
|  | MAmin | 1318 | 183 | 97 | 79 | 378 | 78 | 455 |
| WAD2M | MAmax | 6756 | 1077 | 743 | 1675 | 678 | 1308 | 985 |
|  | MAmean | 4208 | 719 | 370 | 776 | 572 | 760 | 778 |
|  | MAmin | 2437 | 479 | 184 | 176 | 482 | 293 | 639 |
| GLWDv2$_{ISW+P}$ (Inundated and Saturated Wetland + Peatland) | static | 12486 | 1147 | 2156 | 3280 | 878 | 3023 | 1646 |

**Table 3.** Comparison of GIEMS-MC$_{ISW}$ and GIEMS-MC$_{ISW+P}$ surface extents with WAD2M (Zhang et al., 2021b) and GLWDv2 (Lehner
et al., 2024a) datasets, in $10^3$ km$^2$. For dynamic classes, MAmax and MAmin are shown. GLWDv2$_{ISW}$ refers to the sum of GLWDv2 classes
8 to 21, 28, 29, 31, and 32, while GLWDv2$_{ISW+P}$ refers to the sum of GLWDv2 classes 8 to 29, 31 and 32. Regions correspond to the
shapefiles of HydroATLAS database (version 1.0; Linke et al. (2019)).

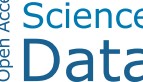

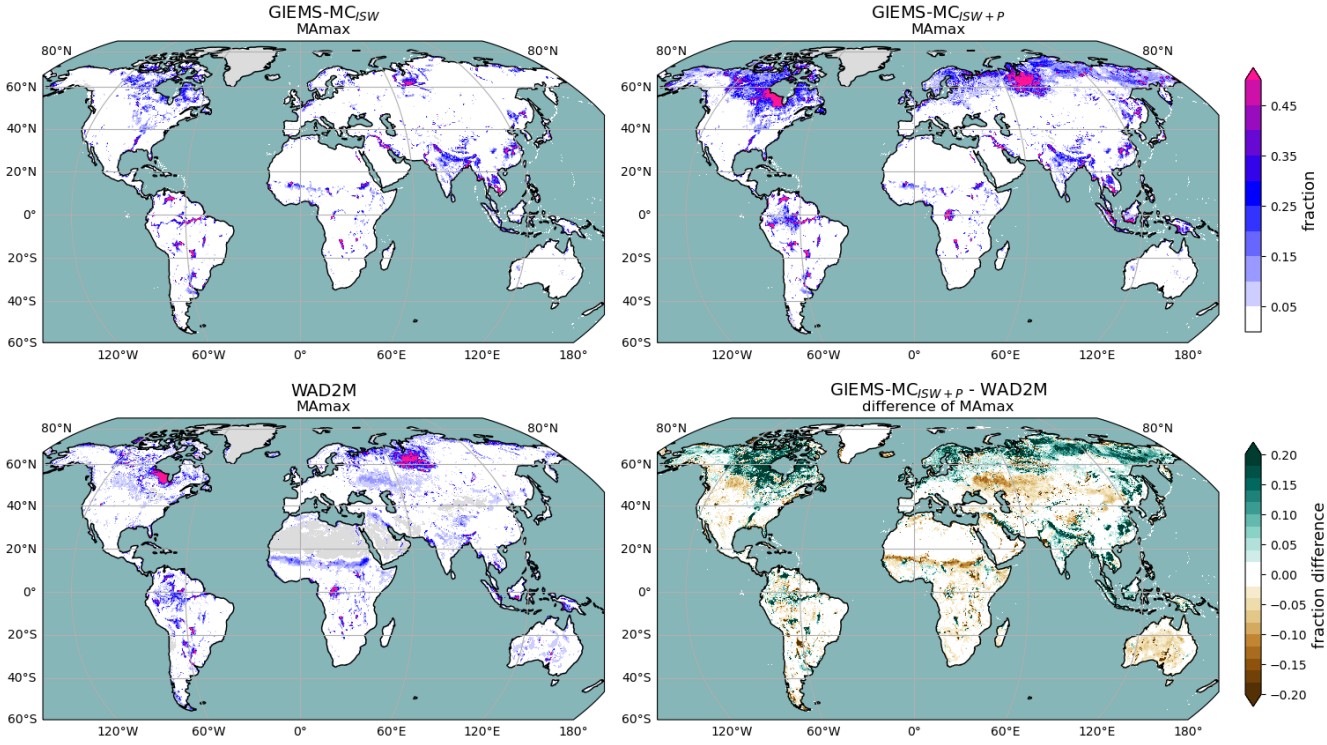

**Figure 2.** Global distribution of the MAmax of GIEMS-MC$_{ISW}$, GIEMS-MC$_{ISW+P}$, and WAD2M (Zhang et al., 2021b), as well as the difference of MAmax from GIEMS-MC$_{ISW+P}$ and WAD2M. Refer to Supplementary Fig. S1 for maps with MAmin.

Figure 3 provides the latitudinal distribution of a) GIEMS-MC variables and b) GIEMS-MC$_{ISW+P}$ against WAD2M. GIEMS-MC$_{ISW}$ shows a relatively uniform distribution across all latitudinal zones, with a peak just south of the equator due to the Amazon basin. The inclusion of peatlands in GIEMS-MC$_{ISW+P}$ increases largely the wetland area in the boreal (>55°N, e.g., the Hudson Bay and the Siberian Low Lands) and tropical (10°S-5°N, e.g., the Congo) bands, leading to a similar distribution as in WAD2M (Fig. 2 and 3.c). Differences between GIEMS-MC$_{ISW+P}$ and WAD2M are observed around 15°N and 10-30°S, due to discrepancies between the SWAMPS and GIEMS-2 methodologies (Pham-Duc et al., 2017; Bernard et al., 2024b), mostly related to desert contamination in SWAMPS (Sahel and Australia on Fig. 2).



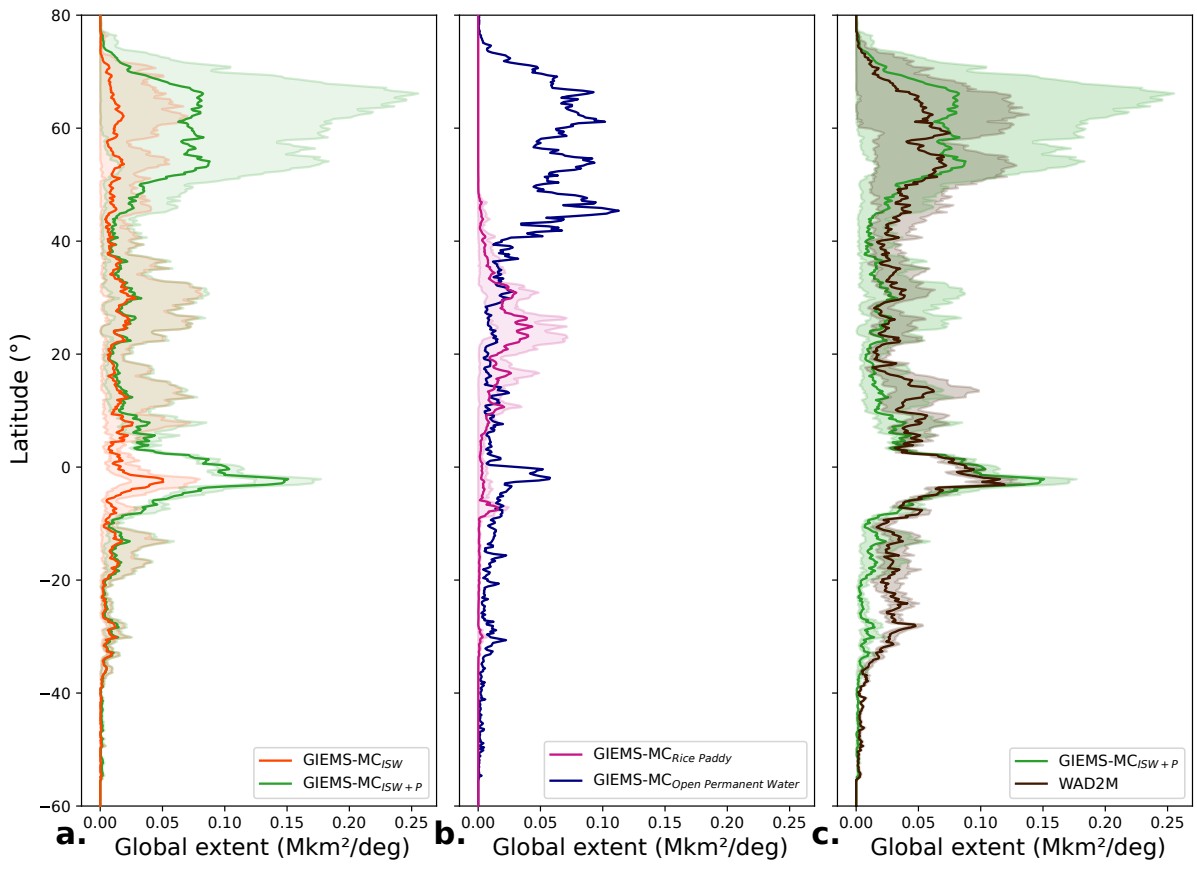

**Figure 3.** Latitudinal distributions of **a.** the GIEMS-MC wetland variables (Inundated and Saturated Wetlands, Inundated and Saturated Wetlands + Peatlands), **b.** GIEMS-MC wetland ancillary variables (Rice Paddy, and the sum of Open Permanent Water), and **c.** GIEMS-MC$_{ISW+P}$ and WAD2M product. For dynamic variables, solid lines represent the MAmean, while colored fillings represent the MAmax-MAmin interval. The extents are given per 1-degree latitudinal bin.

## 4.2 Regional spatial patterns over main basins

GIEMS-MC$_{ISW}$ and GIEMS-MC$_{ISW+P}$ data are analyzed in the following sections over large wetland complexes representing different environments : the Siberian Lowlands, the Sudd, the Amazon and South-East Asia.

300    As expected, peatland addition noticeably amplifies the extent between GIEMS-MC$_{ISW}$ and GIEMS-MC$_{ISW+P}$ over the Ob basin (Western Siberian Low Lands, Fig. 4). WAD2M and GIEMS-MC$_{ISW+P}$ consistently present similar patterns. However, discrepancies occur in the southern part of the Ob basin that can be attributed to different snow filtering between SWAMPS (used for WAD2M) and GIEMS-2 (used for GIEMS-MC).

In the Sudd basin shown in Fig. 5, GIEMS-MC$_{ISW+P}$ extent corresponds essentially to GIEMS-MC$_{ISW}$, indicating minimal
305    presence of peatlands. For comparison, two other products, both derived from Cyclone Global Navigation Satellite System





L-band remote sensing observations, are also shown (Zeiger et al., 2023; Gerlein-Safdi et al., 2021). Gerlein-Safdi et al. (2021) estimates are available for the southern part of the basin (MAmax of 0.27 Mkm$^2$ for 2018-2019), and are much higher than the GIEMS-MC (MAmax of 0.04 Mkm$^2$ for 2018-2019) and WAD2M (MAmax of 0.1 Mkm$^2$ for 2018-2019) estimates. Zeiger et al. (2023) product provides an MAmax of 0.06 over 2018-Aug to 2019-Jul, which is within the GIEMS-MC and WAD2M estimates. While good agreement is observed in the southern part of the basin between the spatial pattern of GIEMS-MC$_{ISW+P}$, WAD2M, and the product of Zeiger et al. (2023), significant disparities emerge between WAD2M and the two products in the northern-east desert region of the Sudd basin, probably due to contamination in the original SWAMPS dataset.

Over the Amazon (Fig. 6), GIEMS-MC$_{ISW}$ fractions are high (>0.5) along the main river channel, while including peatlands adds smaller surfaces, resulting in finer spatial patterns. The resulting GIEMS-MC$_{ISW+P}$ MAmax map closely resembles that of WAD2M.

GIEMS-MC$_{ISW+P}$ and WAD2M agree well in South-East Asia (Fig. 7), with GIEMS-MC$_{ISW+P}$ showing greater peatland extent than WAD2M due to higher peatland areas estimated by GLWDv2 used in GIEMS-MC production than the earlier estimates used in WAD2M.

These findings underline the legacy of the two original microwave-based datasets (GIEMS-2 and SWAMPS) used respectively in GIEMS-MC and WAD2M production, despite the corrections. Indeed, methodological disparities between GIEMS-2 and SWAMPS production may lead to distinct spatial inundation detection patterns, particularly in regions where contamination from ocean, desert, and snow need careful consideration.

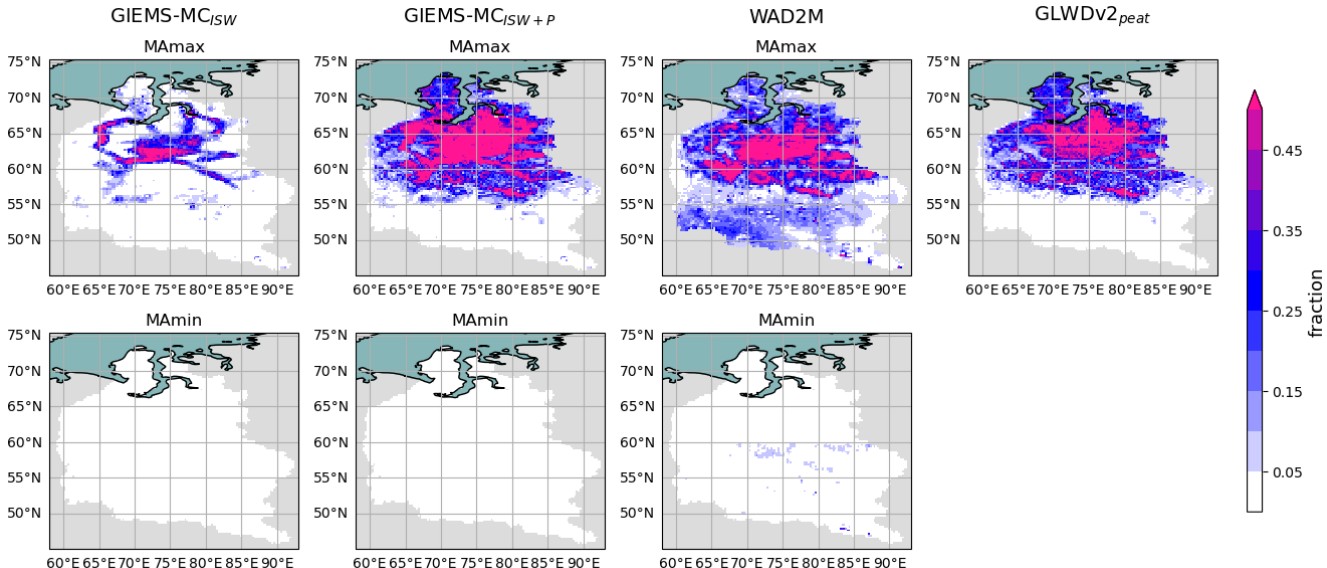

**Figure 4.** MAmax and MAmin maps of GIEMS-MC$_{ISW}$ (1992 to 2020), GIEMS-MC$_{ISW+P}$ (1992 to 2020), and WAD2M (2000 to 2020) over the Ob, as well as GLWDv2 (static) peatland map. Low MAmin are due to the snow mask.



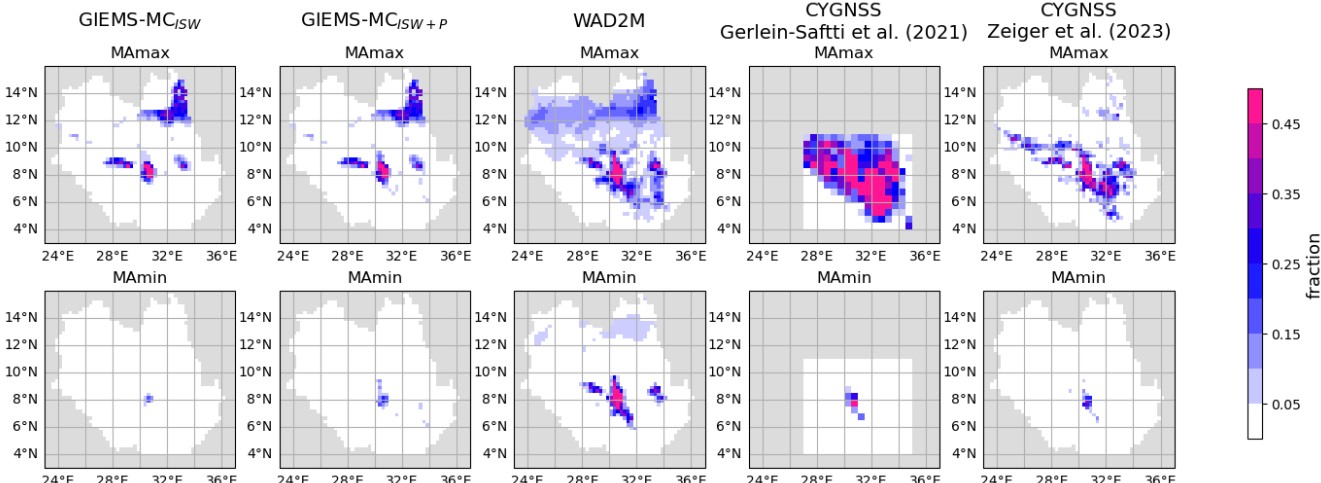

**Figure 5.** MAmax and MAmin maps over the Sudd region of GIEMS-MC$_{ISW}$, GIEMS-MC$_{ISW+P}$ (1992 to 2020), WAD2M (2000 to 2020) and CYGNSS-based estimates from Gerlein-Safdi et al. (2021) (2017-Jun to 2020-Apr) and Zeiger et al. (2023) (2018-Aug to 2020-Jul). The available periods differ between the products.

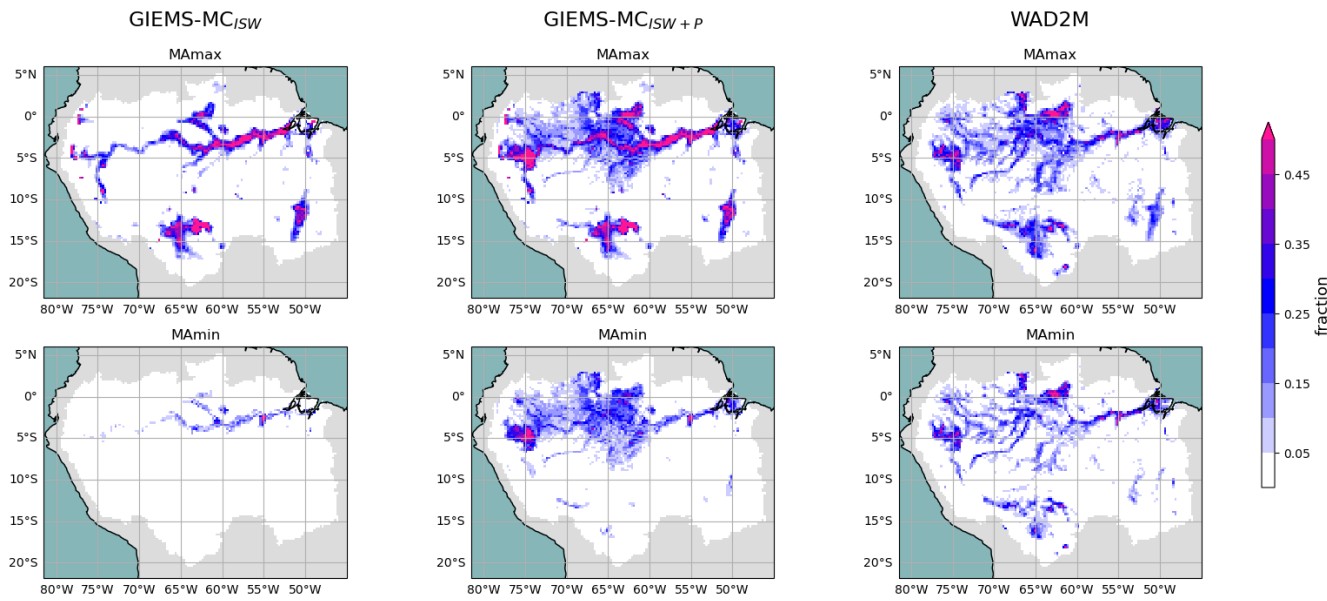

**Figure 6.** MAmax and MAmin maps of GIEMS-MC$_{ISW}$ (1992 to 2020), GIEMS-MC$_{ISW+P}$ (1992 to 2020), and WAD2M (2000 to 2020) over the Amazon basin.





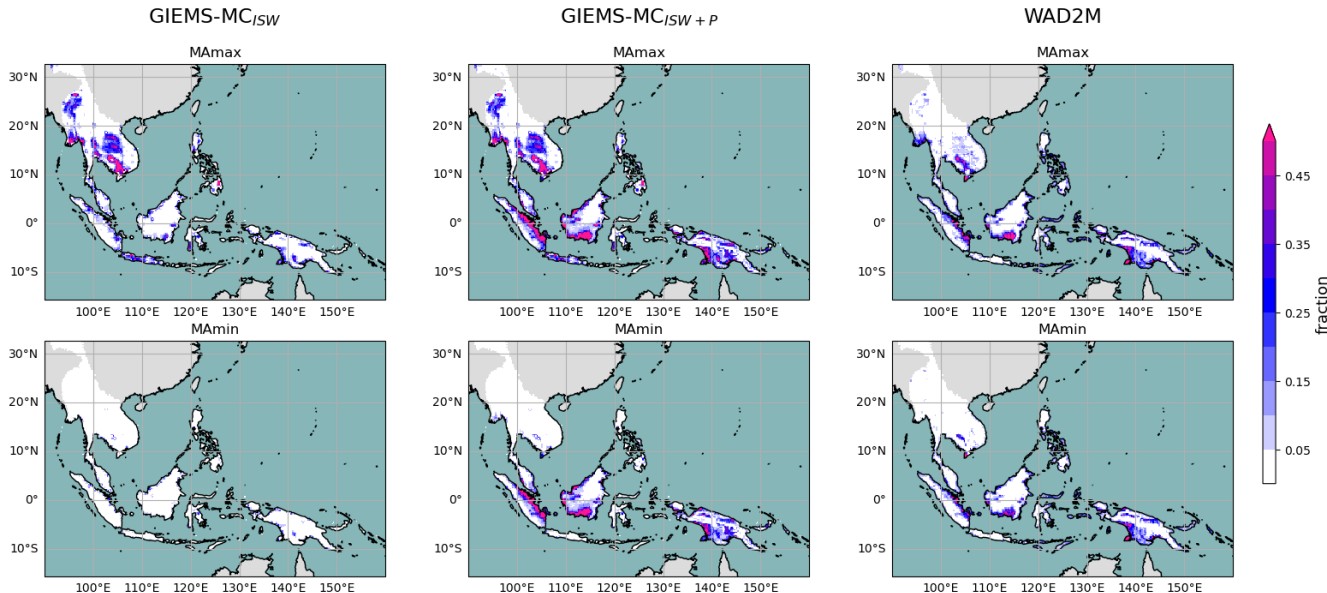

**Figure 7.** MAmax and MAmin maps of GIEMS-MC$_{ISW}$ (1992 to 2020), GIEMS-MC$_{ISW+P}$ (1992 to 2020), and WAD2M (2000 to 2020) over South-East Asia.

### 4.3 Temporal seasonal and inter-annual variations

The temporal dynamics of GIEMS-2 was extensively examined in Prigent et al. (2020) and evaluated in Bernard et al. (2024b), where it was compared with other hydrological observations, including MODIS-derived surface water extent (Frappart et al., 2018; Normandin et al., 2018, 2024), CYGNSS-derived (Zeiger et al., 2023) surface water extent, and river discharge. The evaluation showed that GIEMS-2 reliably captures temporal variations, including seasonality and inter-annual variabilities, even in regions with dense vegetation cover (Fig. 8 and Fig. 9).





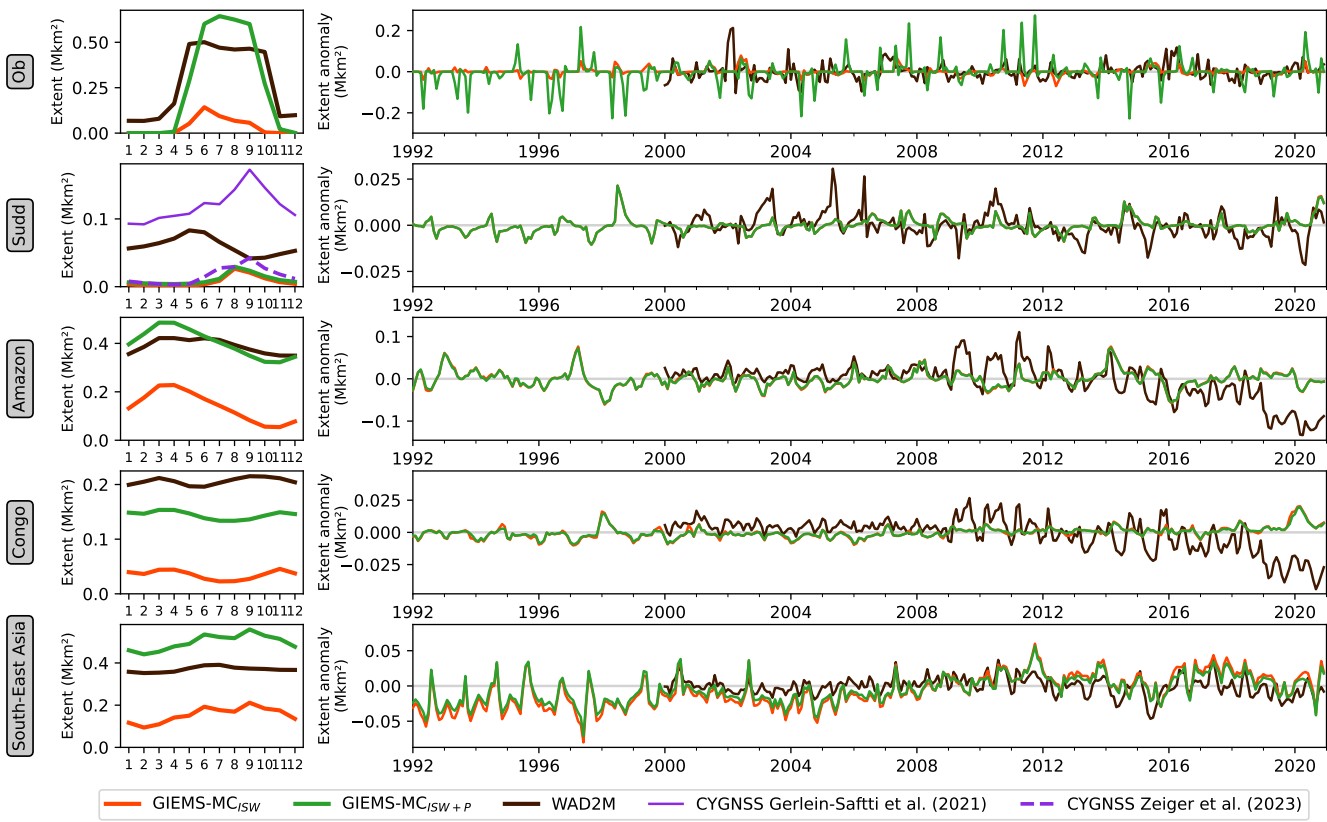

**Figure 8.** Left: Monthly mean seasonal cycle of GIEMS-MC$_{ISW}$, GIEMS-MC$_{ISW+P}$, and WAD2M over different regions. Right: Deseasonalized monthly anomalies of the same three variables. To derive the deseasonalized monthly anomalies, the average monthly seasonal cycle was subtracted from the long term monthly time series. For the Sudd basin seasonality comparison, estimations from Gerlein-Safdi et al. (2021) (2017-06 to 2020-04) and Zeiger et al. (2023) (2018-08 to 2020-07) are added.

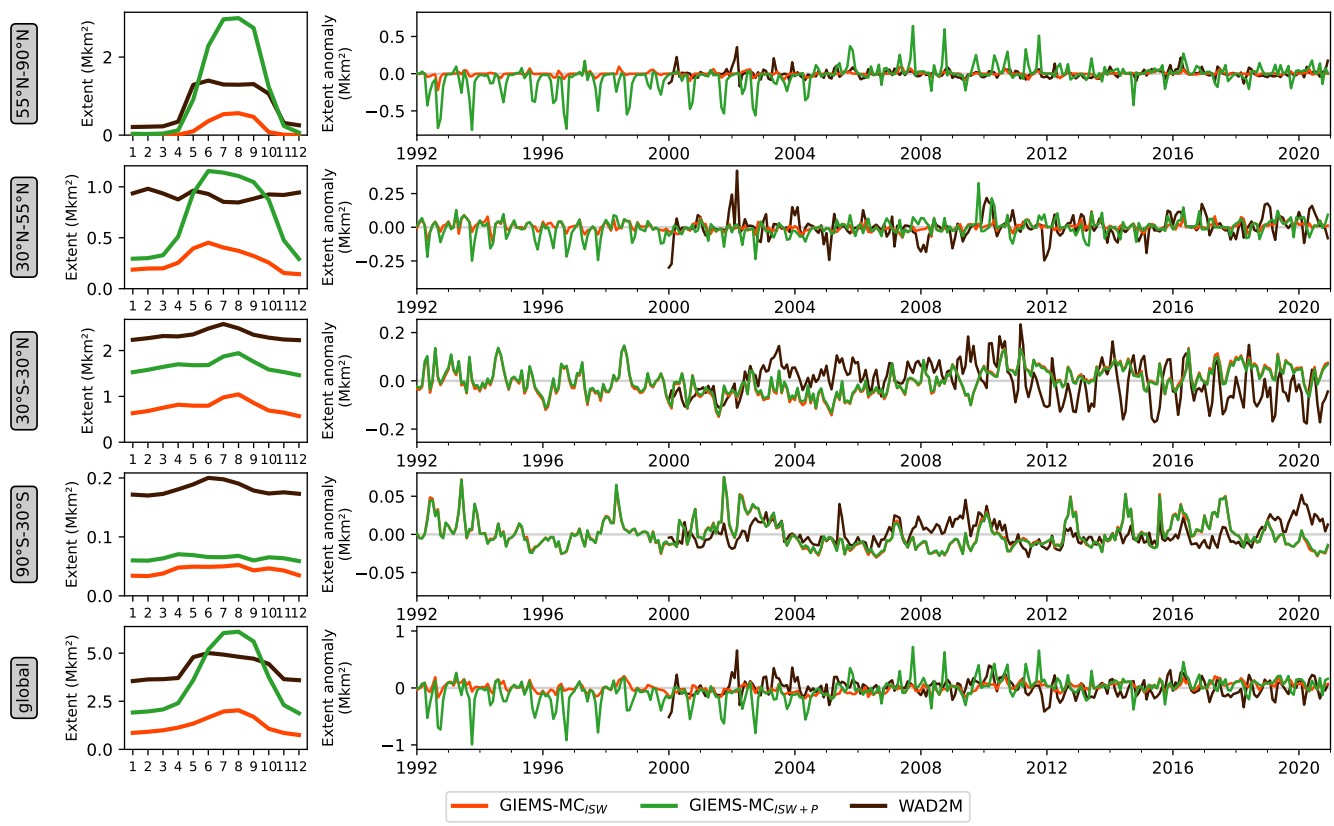

**Figure 9.** Left: Monthly mean seasonal cycle of GIEMS-MC$_{ISW}$, GIEMS-MC$_{ISW+P}$, and WAD2M over different latitudinal bands. Right: Deseasonalized monthly anomalies of the same three variables. To derive the deseasonalized monthly anomalies, the average monthly seasonal cycle was subtracted from the long term monthly time series.

### 4.3.1  Seasonal variations

The 2000-2020 mean seasonality of GIEMS-MC$_{ISW}$, GIEMS-MC$_{ISW+P}$, and WAD2M over the Ob, the Sudd, the Amazon, the Congo, and South-East Asia, are presented in Fig. 8 (left). The seasonal variations of the GIEMS-MC variables are driven by the dynamics of saturated and inundated wetlands (GIEMS-MC$_{ISW}$), with peatlands contributing to an offset effect between GIEMS-MC$_{ISW}$ and GIEMS-MC$_{ISW+P}$. In the Ob, the Amazon, the Congo, and South-East Asia, GIEMS-MC$_{ISW+P}$ exhibits comparable magnitudes than WAD2M. In the Sudd region, WAD2M shows distinct seasonality than GIEMS-MC and the two

CYGNSS derived products, probably due to the SWAMPS artifacts in desert regions mentioned above.

     Across latitudinal bands, the global seasonality of GIEMS-MC$_{ISW}$ and GIEMS-MC$_{ISW+P}$ are mainly driven by the boreal and temperate northern regions, due to snow cover changes (Fig. 9, left). However, notable differences in terms of seasonal cycle between GIEMS-MC$_{ISW+P}$ and WAD2M exist over the mid to high latitudes. Indeed, larger peatland surfaces are included in GIEMS-MC than in WAD2M (higher amplitude) and the more widespread snow masking in GIEMS-MC in the temperate

zone leads to a stronger seasonal cycle compared to WAD2M. The seasonal cycles over the tropics and southern hemisphere are more similar between GIEMS-MC$_{ISW+P}$ and WAD2M, but surface extents are larger in WAD2M, predominantly due to desert and ocean contamination in SWAMPS, as discussed in Sect. 4.1.

### 4.3.2   Inter-annual variations & trends

Figure 8 (right) shows the deseasonalized monthly anomalies of GIEMS-MC$_{ISW}$, GIEMS-MC$_{ISW+P}$, and WAD2M over differ-

ent regions, while Fig. 9 (right) corresponds to latitudinal bands. The reference seasonality period subtracted is the 2000-2020 seasonal average. As expected, GIEMS-MC$_{ISW}$ and GIEMS-MC$_{ISW+P}$ have the same anomalies for latitudes below 30°N because the temporal dynamics comes from the inundated and saturated wetlands, and not from the static peatland map. For northern temperate and boreal areas, the snow cover also imposes a seasonality on peatlands, which explains larger anomalies of GIEMS-MC$_{ISW+P}$ than in GIEMS-MC$_{ISW}$. For GIEMS-MC$_{ISW+P}$ over the boreal region (55°N-90°N), a positive trend is

detected over May and June (+10 $10^3$ km$^2$ yr$^{-1}$) and September and October (+24 $10^3$ km$^2$ yr$^{-1}$) months that can possibly be attributed to earlier snow melt and delayed snow cover arrival.

     No long-term trends were found at regional scales in GIEMS-MC$_{ISW}$, except for South-East Asia, where a small positive trend was found (+1.7 $10^3$ km$^2$ yr$^{-1}$, i.e. ∼+50 $10^3$ km$^2$ for 30 yr). This is likely mostly due to the increasing trend in rice paddies, which is not taken into account because only the MIRCA2000 climatology is used, and new rice paddies over the

years are then aliased to wetlands over time (see Discussion Sect.5.2.2).

     An abrupt change in WAD2M inter-annual variability amplitude occur over the Amazon and Congo basins in 2009, attributed to a change in one of the satellite data used in SWAMPS, together with a decreasing trend also found in the SWAMPS data (Fig. 8). Due to these problems in SWAMPS, the inter-annual variability of WAD2M should be considered with caution, and makes time-series comparison with GIEMS-MC$_{ISW+P}$ difficult over these regions.

## 5   Discussion

The production of GIEMS-MC involves seven steps, each of which contributes to the transition from inundation time series to wetland map time series, but also to the uncertainties of the final product. It has been estimated that the GIEMS product possibly underestimates surface water areas by less than 10% (Prigent et al., 2007). This value can be used as an order of magnitude of the uncertainty in GIEMS-2, although methodological improvements have been made between GIEMS and GIEMS-2 (Prigent

et al., 2020). This is also likely a realistic approximation for the GIEMS-MC$_{ISW}$ error, as it uses mainly GIEMS-2 information. A quantification of the uncertainties of the GIEMS-MC variables would require a deeper knowledge of the measurement and detection uncertainties of all the products used, some of which are not calculated in the original source, which is beyond the scope of this study. However, it is possible to quantify the influence of each step in the GIEMS-MC procedure and to study the sensitivity of the results to the processing choices.



## 5.1 Quantification of the influence of each process step on the GIEMS-MC global extents

Table 4 shows the influence of the successive steps in terms of MAmax, MAmean and MAmin. The removal of open permanent water (Step 4.), as well as the subtraction of rice paddies (Step 5.), have a significant impact on the global extent, both resulting in a subtraction of -2.34 Mkm$^2$ on MAmax and -1.16 Mkm$^2$ on MAmean. Coastal cleaning also has a large influence on the reduction of the area: -2.60 Mkm$^2$ in MAmax and -1.60 Mkm$^2$ in MAmean, and especially on MAmin, which decreases from 1.17 Mkm$^2$to 0.26 Mkm$^2$. In fact, the coastal region has an artificially high minimum value due to ocean contamination before this cleaning step. Finally, the addition of peatlands turns out to be an extremely significant step in terms of surface area for GIEMS-MC$_{ISW+P}$, with large increases observed in both MAmax (+3.94 Mkm$^2$) and MAmean (+2.27 Mkm$^2$) extents.

| Step | Step description | MAmax | MAmean | MAmin |
|------|------------------|-------|--------|-------|
| 0 | GIEMS-2 revised version | 9.02 | 4.15 | 1.63 |
| 1 to 3 | After masking oceans, snow and urban areas | 8.83 | 4.03 | 1.58 |
| 4 | After subtracting open permanent water | 7.10 | 3.20 | 1.30 |
| 5 | After subtracting rice paddies | 6.49 | 2.87 | 1.17 |
| 6 | After cleaning coasts (GIEMS-MC$_{ISW}$) | **3.89** | **1.27** | **0.26** |
| 7 | After adding peatlands (GIEMS-MC$_{ISW+P}$) | **7.83** | **3.54** | **1.32** |

**Table 4.** Global MAmax, MAmean and MAmin in Mkm$^2$ after each step of GIEMS-MC production. It should be noted that snow and oceans were already set to zero fraction in GIEMS-2, and that the snow mask is applied to all wetlands, including peatlands, and is then responsible for the low MAmin values.

## 5.2 Sensitivity to the GIEMS-MC procedure

The three critical stages in the production of GIEMS-MC are further discussed in this section, along with the use of a snow mask.

### 5.2.1 Coastal processing

In the Methods Sect. 3.1.6, we chose to apply a cleaning procedure to coastal pixels located within 50 km of the coast. Figure 10.a shows the global mean seasonality of GIEMS-MC$_{ISW}$ and GIEMS-MC$_{ISW+P}$ when considering coastal bands ranging between 0 and 100 km from the coast to be processed using GLWDv2 information, following the methodology in Sect. 3. After removing the pixels with more than 10% ocean, the cleaning of the coastal band up to 30 km from the coast reduces GIEMS-MC$_{ISW}$ MAmax by 0.7 Mkm$^2$. Cleaning also the 30-50 km band reduces it further by 0.5 Mkm$^2$, while the 50-70 km (-0.12 Mkm$^2$) and 70-100 km (-0.08 Mkm$^2$) bands have smaller effects. The cleaning over 50 km is consistent with our technical understanding of the contamination.



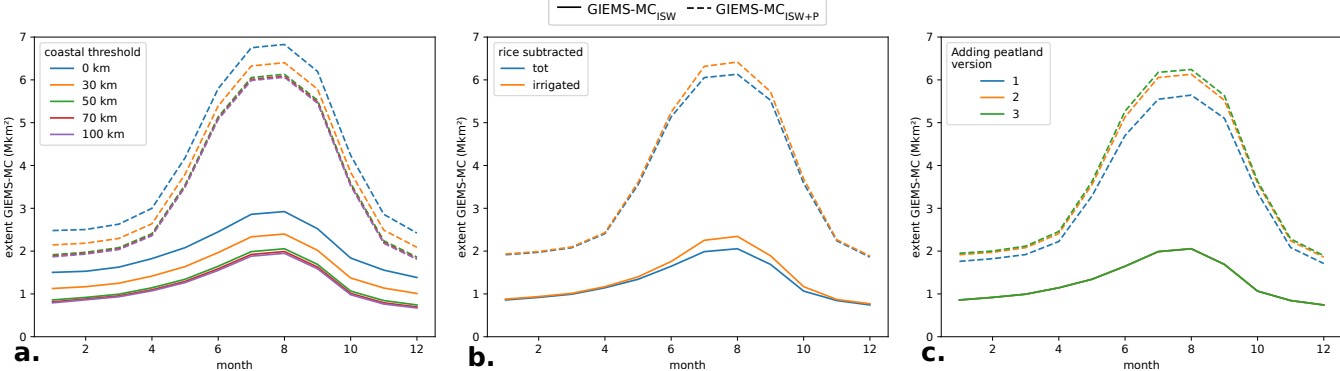

**Figure 10.** Sensitivity of GIEMS-MC averaged seasonality to the different steps of the production procedure : **a.** the coastal threshold for coastal cleaning, **b.** the rice procedure, and **c.** the way peatlands are added. Solid lines represent the extent of GIEMS-MC$_{ISW}$, while dashed lines represent GIEMS-MC$_{ISW+P}$. Colors show the different GIEMS-MC treatments.

### 5.2.2 Rice subtraction

An issue concerning rice in GIEMS-MC production stems from the classification used in the MIRCA2000 dataset, which separates rice paddies into irrigated and rainfed types. Irrigated paddies are typically fully inundated at least part of the year. Rainfed paddies have variable levels of submergence, with approximately 80% inundated and 20% remaining upland paddies (Maclean et al., 2013). Under these conditions, the upland rice (9% of total rice paddies area ; Maclean et al. (2013)) should not be subtracted from GIEMS-2 in the GIEMS-MC processing steps, if we could distinguish upland rice from the rest of

the rainfed paddies. To explore this, we attempted a classification based on topographic information to separate inundated from non-inundated within the rainfed rice category. The resulting maps, shown in Supplementary Table 3, lead to surface extent inconsistent with FAO statistics per country (FAO, 2002). In the absence of any reliable distinction possibility, the total rice paddies were subtracted, acknowledging that the subtracted area might be overestimated by about 9%. Note that this light overestimation of subtracted rice paddies might be counterbalanced by the fact that MIRCA2000 areas estimates are

underestimated when compared to FAOSTAT (Fig. 11).

In WAD2M production, the MIRCA2000 is also used to differentiate rice paddies from wetlands, but only irrigated paddy class is subtracted (Zhang et al., 2021b). To evaluate the impact of rice handling in GIEMS-MC, Fig. 10.b shows global seasonality of GIEMS-MC$_{ISW}$ and GIEMS-MC$_{ISW+P}$ if only irrigated rice paddies are subtracted, or if both irrigated and rainfed are subtracted. The difference occurs mainly between June and October, in the Northern Hemisphere summer, corresponding

to a difference of 0.25 Mkm$^2$ in terms of MAmax (6% of GIEMS-MC$_{ISW}$ and 3% GIEMS-MC$_{ISW+P}$ MAmax). While this has a moderate influence on global extent, this difference can be important in rice cultivating countries, e.g., a difference of 30% in GIEMS-MC$_{ISW}$ MAmax over India depending on the type of subtraction used. For GIEMS-MC$_{ISW+P}$, as the total surfaces are higher, the influence of rice paddy subtraction is proportionally less important.

Finally, subtracting the MIRCA2000 climatology in the GIEMS-MC processing and not taking into account the inter-

annual variation of some rice paddies over the period 1992-2020 can lead to misclassification of rice paddies as wetlands.





The MIRCA2000 product is compared in Fig. 11 with the estimates from the Food and Agriculture Organization (FAO) of the United Nations estimates FAOSTAT (https://www.fao.org/faostat/en/#data/QCL, access 30/06/2023). FAOSTAT is widely used for global estimates of methane emissions from rice paddies, notably in the Emissions Database for Global Atmospheric Research (EDGAR ; Janssens-Maenhout et al. (2019)). The cropland area of rice paddies is increasing in South-East Asia,

with FAOSTAT estimating $+60\ 10^3$ km$^2$ between 1992 and 2020 in this region, which corresponds to the increasing trend of $\sim +50\ 10^3$ km$^2$ in GIEMS-MC$_{ISW}$ over this period (Sect. 4.3.2).

MIRCA2000 (MAmax of 1.25 Mkm$^2$) presents smaller rice paddies extent than FAOSTAT (1.47 Mkm$^2$ in 1992 to 1.64 Mkm$^2$ in 2020). In GLWDv2, the map from Salmon et al. (2015) is used as primary information but undergoes numerous corrections related to artifacts in the product, including double-checking information using the RiceAtlas (Laborte et al., 2017). This lead to

a static map of rice paddies of 1.2 Mkm$^2$, close to MIRCA2000 MAmax estimates. Then, various inventories of anthropogenic methane emissions that are accounting for rice methane emissions are not using the same maps for rice paddies, which can lead to mismatches across the estimates (surfaces double counting or miscounting). Efforts to use similar compatible rice maps between the two research communities would greatly improve the consistency of wetland time series, and then the methane emission estimates. A dynamic map that accurately reflects the temporal variation of inundated rice paddies would better meet

the needs of remote sensing wetland mapping. This approach would address the limitations of existing classifications, such as MIRCA's irrigated/rainfed or FAO's yearly irrigated/rainfed/upland categories, which do not adequately address the specific needs of the community.

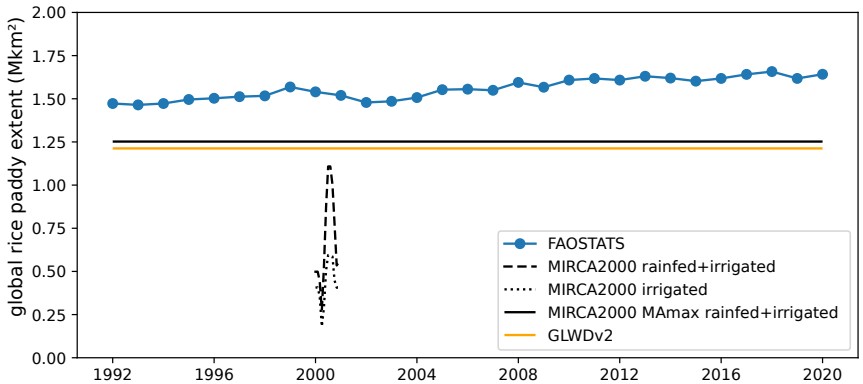

**Figure 11.** Rice paddy surface extent estimations from MIRCA2000 (Portmann et al., 2010), FAOSTAT (https://www.fao.org/faostat/en/#data/QCL), and GLWDv2 (Lehner et al., 2024a). MIRCA2000 MAmax is higher than the maximum of the MIRCA2000 seasonality plot because not all pixels have their maximum in the same month.

### 5.2.3  Peatland integration

Peatlands contribute to more than a half of areas in GIEMS-MC$_{ISW+P}$ (Table 4), and depends highly on the GLWDv2 peatland

map used here.



Most of the peatlands are not saturated or inundated areas, although some can have their water table above the peat surface intermittently (Lourenco et al., 2023). A large part of peatlands are then not detected by GIEMS-2. Three ways of integrating GLWDv2 peatlands were tested to assess how sensitive the peatlands integration method is. For each pixel $i$, we did the following:

1. Set $GLWDv2_{peat}$ fraction as the minimum of $GIEMS\text{-}MC_{ISW+P}$, minimizing $GIEMS\text{-}MC_{ISW+P}$ areas:

    if $GIEMS\text{-}MC_{ISW i} < GLWDv2_{peat i}$, then $GIEMS\text{-}MC_{ISW+P i} = GLWDv2_{peat i}$.

2. Attempt to add only the peatlands not detected by GIEMS-2 as described in Methods Sect. 3.1.7:

    $GIEMS\text{-}MC_{ISW+P i} = GIEMS\text{-}MC_{ISW i} + pos[\ GLWDv2_{peat i} - pos(\ GIEMS\text{-}MC_{ISW i} - GLWDv2_{ISW+P i}\ )\ ]$

3. Add all GLWDv2 peatland, maximizing $GIEMS\text{-}MC_{ISW+P}$ areas:

    $GIEMS\text{-}MC_{ISW+P i} = GIEMS\text{-}MC_{ISW i} + GLWDv2_{peat i}$.

The effect of these three peatland integration approaches on $GIEMS\text{-}MC_{ISW+P}$ extent are represented in Fig. 10.c. A difference of 0.85 Mkm$^2$ (11%) is found for $GIEMS\text{-}MC_{ISW+P}$ MAmax between the two extreme approaches 1 and 3. Approach 1 likely underestimates peatland integration, as some pixels can contain both inundated or saturated wetlands and peatlands. Approach 3 likely overestimates peatland integration, as some peatlands (inundated and saturated) should be detected in GIEMS-

2. The method 2 appears as a sensible consensus, but this approach also likely overestimates peatland surfaces, as GLWDv2 wetland categories, used to discriminate detected and non-detected peatlands by GIEMS-2 (see Sect. 3.1.7), is a long term maximum.

### 5.2.4 Snow-covered pixel masking

Due to the influence of snow on passive microwave observations, snow-covered pixels are masked in the estimation of GIEMS-

2 and GIEMS-MC inundation fractions (see 3.1.2). This masking prevents models from accounting for methane emissions from snow-covered areas. However, cold-season methane fluxes in arctic peatlands and tundra have been shown to contribute between 25% and 50% of the annual local fluxes (Bao et al., 2021; Ito et al., 2023; Mastepanov et al., 2008; Rößger et al., 2022; Zona et al., 2016). Therefore, the sensitivity of microwave remote sensing to snow is a limitation in boreal regions. Nevertheless, the boreal zones are estimated to contribute only about 5% of annual global wetland and inland freshwater

455
emissions using a top-down approach, and about 10% using a bottom-up approach (Saunois et al., 2024), with only up to half of these boreal emissions potentially occurring during the cold season. Consequently, the exclusion of snow-covered areas is likely to add only a few percents of uncertainty to global methane emissions from wetlands and inland waters.

## 6 Perspectives

Several key areas for future improvement of the GIEMS-MC production process were identified. First, taking into account

460
the inter-annual variations of rice paddies would help improve the accuracy and in particular the long-term trend of GIEMS-MC. Ideally, these estimates should be consistent with those used in anthropogenic greenhouse gas emission inventories such



as FAOSTAT. Secondly, a better distinction between inundated and dry peatlands would allow a more accurate integration of peatlands into GIEMS-MC$_{ISW+P}$. New satellite data, such as the 2022-launched Surface Water and Ocean Topography (SWOT) with its Ka-band Radar Interferometer, hold promise for the monitoring of continental surface water area and height at high spatial resolution and temporal sampling (Neeck et al., 2012; Pedinotti et al., 2014; Biancamaria et al., 2016; Prigent et al., 2016; Peral and Esteban-Fernandez, 2018). In particular, either high resolution SWOT data or water table depth from a hydrological model combined with the 500 m GLWDv2 data could help to better distinguish the inundated from the non-inundated peatlands to improve the integration of non-inundated peatlands in GIEMS-MC$_{ISW+P}$. In addition, the upcoming NASA-ISRO Synthetic Aperture Radar (NISAR), scheduled for launch in 2025, will provide high-resolution (below 7 m) observations in L and S bands (Chuang et al., 2016; Adeli et al., 2021). These frequencies are particularly advantageous for mapping of sub-canopy inundation in forested wetlands, as they penetrate vegetation more effectively than the Ka band used for SWOT.

GIEMS-MC dynamics is derived from GIEMS-2, which provides valuable insights into global water surface dynamics with seamless time series of surface water extent. The continuity of GIEMS-2 production holds the potential to extend the temporal coverage of the GIEMS-MC maps. However, no new SSMIS instrument is planned to be launched when the current instruments (F15 to F18) are decommissioned. Adaptations to the GIEMS-2 process, such as the incorporation of Advanced Microwave Scanning Radiometer (AMSR) data, will be required to extend the observation period despite critical changes in satellite overpassing local time and spatial resolution. Other passive microwave future missions are expected to cover all or part of the SSMIS microwave frequency range, such as the MicroWave Imager (MetOp-SG, D'Addio et al. (2014)) or the Copernicus Imaging Microwave Radiometer (CIMR, Vanin et al. (2020)), offering alternative data sources for GIEMS-MC production but also requiring adjustment to the methodology. In addition, plans to increase the temporal resolution of GIEMS-2 to a 10-day data record could provide a more detailed understanding of wetland dynamics over time.

# 7 Conclusion

Despite numerous advances in methane measurements and modeling, the extent of wetlands remain a key gap. Here, GIEMS-2 product was combined with other information to produce GIEMS-MethaneCentric (GIEMS-MC), a dataset containing spatially and dynamically consistent maps of different methane-emitting aquatic ecosystems. In particular, GLWDv2 dataset enables the separation of open water surfaces (lakes, rivers, reservoirs) in GIEMS-2, as well as the addition of peatlands not detected by microwaves satellite observations used in GIEMS-2 production. Rice paddies are identified using the MIRCA2000 product. Updated coastal zone filtering improves on the previous complete masking in the distributed version of GIEMS-2.

GIEMS-MC provides two harmonized times series maps at 0.25°x0.25° and monthly time step of wetland surfaces from 1992 to 2020: one representing inundated and saturated wetlands, and the other covering all wetlands, including peatlands. In addition, GIEMS-MC provides consistent maps of rice paddies and categories of open permanent water. Information on the dominant vegetation type and wetland type per pixel is also provided. This comprehensive database will hopefully set a new standard for harmonizing and consistently mapping methane emission from the different aquatic ecosystems.

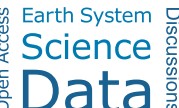

495   *Data availability.*   GIEMS-MC dataset in NetCDF format and its documentation are available at https://zenodo.org/records/13919645 (Bernard et al., 2024a).

*Author contributions.*   J.B., C.P., C.J., M.S. and E.F-C. conceived the main ideas of this study. C.J. and C.P. developped and produced the GIEMS-2 data. B.L. provided GLWDv2 product. J.B., C.J., and C.P. built the database and performed the numerical analyses. J.B. drafted the manuscript with input from C.P., M.S, and E.F-C. All authors provided critical feedbacks and expertise on the manuscript.

500   *Competing interests.*   The authors declare no competing interests.

*Acknowledgements.*   Juliette Bernard is funded by a PhD grant from the Institut National des Sciences de l'Univers (INSU) of the Centre National de la Recherche Scientifique (CNRS). Partial funding has been provided by the ESA CCI RECAPP2 project (4000123002/18/INB), and a preliminary version of this database has been developed under that contract. The Agence National de la Recherche also provided support through the project Advanced Methane Budget through Multi-constraints and Multi-data streams Modelling (AMB-M3 - ANR-21-
505   CE01-0030).





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
