# Peer review of "The GIEMS-MethaneCentric database: a dynamic and comprehensive global product of methane-emitting aquatic areas"

_Earth System Science Data, 2024_

## Author Comment (AC1)

**Replies to Referee 1**

Bernard and co-authors' manuscript is a concise, clear description of an updated version of the well-known GIEMS inundation product called GIEMS-MethaneCentric (GIEMS-MC). Modifications include separating the areas of open water using the Global Lakes and Wetlands Database version 2 (GLWDv2) and rice paddies from wetland areas, and using filters for coastal zones to avoid ocean artifacts and for regions with snow cover. GIEMS-MC spans the period 1992-2020 on a monthly timescale at 0.25° x 0.25° spatial resolution, and includes one time-series of flooded and saturated wetlands and one for wetlands and peatlands, plus static areas of lakes, rivers and reservoirs, seasonal rice paddies and dominant vegetation. The updated product is compared with the Wetland Area and Dynamics for Methane Modeling (WAD2M) product globally. Regional comparisons are also done for the Siberian Lowlands, the Sudd, the Amazon and South-East Asia. GIEMS-MC is likely to be used by global and regional modeling efforts and should improve estimates of temporal variations in wetland emissions of methane.

Thank you for reviewing this manuscript and for your suggestions that have improved this study. Below you will find point-by-point responses to your comments, together with the corresponding changes made to the new version of the manuscript.

Although the authors state that quantification of the uncertainties of the GIEMS-MC variables is beyond the scope of this study, several aspects of the approach and its validation are in need of further information.

Though the remote sensing basis and limitations of GIEMS and GIEMS-2 are described in a series of publications, a brief summary of these two items would benefit readers of this report.

The description of GIEMS and GIEMS-2 have been modified to be more explicit about these points (lines 102-128 of the track changes document):

*"The GIEMS-2 dataset, spanning 1992 to 2015 and extended to 2020 in this study, provides monthly global maps of surface water extent with a spatial resolution of 0.25°x0.25° (Prigent et al., 2020). GIEMS-2 includes all continental water surfaces, such as wetlands, rice paddies, rivers, reservoirs, and lakes, but excludes large lakes > 15 000 km². The original GIEMS-1 methodology (Prigent et al., 2001, 2007; Papa et al., 2006; Prigent et al., 2007; Papa et al., 2008, 2010) and the GIEMS-2 algorithm (Prigent et al., 2020; Bernard et al., 2024b) have been extensively evaluated against other observational products, demonstrating robust capture of seasonal and inter-annual variability, despite uncertainties in wetland extent between currently available products. GIEMS-2 relies primarily on inter-calibrated passive microwave observations from the SSM/I and SSMIS satellites (Fennig et al., 2020) at frequencies from 19 to 85 GHz. The microwave signal is influenced by atmospheric conditions (e.g., water vapor, clouds) and surface variables (e.g., surface temperature, vegetation, snow). Microwave observations are complemented by ancillary information other to limit these artifacts, as described in Prigent et al., (2020). Specifically, active microwave satellite data and Normalised Difference Vegetation Index (NDVI) derived from visible and near-infrared measurements are used to characterize vegetation and mitigate its influence on the passive microwave signal. Since snow*

*contamination prevents the calculation of surface water, snow-covered pixels (as defined by meteorological reanalysis), are assigned a water fraction of 0 in the previous studies and in the distributed GIEMS-2 product. Passive microwaves are sensitive to the presence of water, including estuarine and offshore marine waters. To avoid misinterpretation of the data, coastal pixels were filtered out in the distributed GIEMS-2 product, leading to a possible underestimation of inundated surface extent in the coastal areas. Here we use an unfiltered version of GIEMS-2 in which coastal regions are not excluded, in order to improve the coastal cleaning process during the production of GIEMS-MC based on GLWDv2, as described in Sect. 3."*

The comparisons of WAD2M and GIEMS-MC are primary illustrated in coarse global figures and in a table. Statistical analyses of global and regional differences between these products is needed. The figures showing temporal variability also need quantitative analysis and comparison with other products, when possible.

In order to do statistical analysis of WAD2M and GIEMS-MC spatial patterns, we calculated spatial correlation coefficients between the two datasets, in terms of MAmax and MAmin, at continental and basin scales. Spatial correlation is defined as the pixel-by-pixel correlation between two maps, considering a specific unmasked region.

| | Correlation of MAmax | | Correlation of MAmin |
|---|---|---|---|
| | **WAD2M** | **BAWLD** | **WAD2M** |
| **Global** | 0.61 | NaN | 0.59 |
| Africa | 0.70 | NaN | 0.76 |
| Europe+Siberia | 0.62 | NaN | 0.12 |
| Asia | 0.37 | NaN | 0.40 |
| Oceania | 0.63 | NaN | 0.62 |
| South America | 0.73 | NaN | 0.66 |
| North America | 0.61 | NaN | 0.49 |
| Sudd | 0.71 | NaN | 0.76 |
| Ob | 0.80 | 0.74 | -0.01 |
| Amazon | 0.77 | NaN | 0.71 |
| Congo | 0.89 | NaN | 0.88 |
| South East Asia | 0.67 | NaN | 0.72 |

We included global and basin metrics into the manuscript text in relevant sections 4.1 and 4.2, along with its definition line 288 of the Track change manuscript. We also added the MAmax and MAmin values of the different datasets over selected basins directly on the Figures (4 to 7), and added it in the relevant paragraphs to use more quantitative metrics.

We do not include more comparison with other surface water products, as we are here interested in wetland products. The extended GIEMS-2 dataset, which detects all surface waters, was compared over 10 basins to other surface water extent products (CYGNSS,

MODIS) and river discharge time series in Bernard et al., 2024 (https://doi.org/10.3389/frsen.2024.1399234). This is mentioned in the manuscript lines 111-114 of the track changes document.

The use of GLWDv2 to separate lake, river and reservoir areas is reasonable for many regions, but not for the widespread and extensive floodplains of tropical and northern rivers because open water areas in these systems vary considerably on a seasonal timescale. A discussion of the consequences of this issue should be added.

We do understand your concern about floodplains that have a seasonal variability not represented in the static maps depicted in GLWDv2. However, only permanent open waters (lakes, rivers*, reservoir, estuaries, and deltas) are removed (separated) from GIEMS-2 using GLWD2. Floodplain ecosystems are included in other GLWDv2 classes (10 to 15) and are therefore retained in GIEMS-2 when constructing GIEMS-MC. This has been explicitly added in the manuscript (lines 215 to 216 of the track changes document):

*"Note that the floodplain ecosystems are included in other GLWDv2 classes (10 to 15) and are therefore retained in GIEMS-2 when constructing GIEMS-MC."*

*Rivers in GLWDv2 are mainly derived from the Global River Width from Landsat (GRWL) dataset (Allen and Pavelsky, 2018), where they are derived during months of average discharge (mean discharge ± one standard deviation, supplement of Allen and Pavelsky, 2018).

The regional comparison for the Amazon basin should mention the evaluation of several inundation products published in Fleischmann et al. (2022). How much inundation occurs in the Amazon River basin? Remote Sensing of Environment. 278, 113099. doi.org/10.1016/j.rse.2022.113099. Given the limitations of the sensors used by GIEMS and the well-validated results from synthetic aperture radar in seasonally inundated forests, this publication provides valuable information about some of the uncertainties inherent in the GIEMS products.

We added a reference to this study (lines 337 to 343 of the track changes document), in particular the extent found by the SAR estimates, highlighting also that wetland and inundated extent are different:

*"Over the Amazon (Fig. 6), GIEMS-MC$_{ISW}$ fractions are high (>0.5) along the main river channel, while including peatlands adds smaller surfaces along smaller secondary channels, resulting in finer spatial patterns and higher MAmax (0.31 to 0.56 Mkm2). The resulting GIEMS-MC$_{ISW+P}$ MAmax map closely resembles that of WAD2M (spatial correlation coefficient of 0.77), with a slightly higher MAmax for WAD2M (0.47 Mkm²). GIEMS-MC$_{ISW}$ extent can be potentially underestimated in this basin, as Fleischmann et al. (2022) found that GIEMS-2 estimates of inundation were likely slightly underestimated compared to higher resolution remote sensing estimates based on Synthetic Aperture Radar (SAR), with Chapman et al. (2015) finding +4% and Rosenqvist et al. (2020) +36% compared to GIEMS-2 long term maximum inundation."*

The Boreal–Arctic Wetland and Lake Dataset (BAWLD) would seem appropriate for comparison in northern regions.

We thank the reviewer for this suggestion. We added BAWLD in the Figure comparison over the Ob basin (Fig. 4), and its comparison within the text (lines 322 to 326 of the track changes document):

*"The Boreal-Arctic Wetland and Lake Dataset (BAWLD) is also shown for comparison, as it covers the upper part of the basin (Olefeldt et al., 2021b, a). The BAWLD wetland fraction, including peatland classes, exhibits patterns similar to GIEMS-MC$_{ISW+P}$ MAmax, with a spatial correlation coefficient of 0.74. However, BAWLD covers a slightly larger area (+6%) compared to the GIEMS-MC$_{ISW+P}$ MAmax when comparing the common area covered by the two datasets."*

Given that peatlands add a large area to the GIEMS-MC results, an evaluation of the veracity of the peatland products is needed. For example, the approach used by Gumbricht et al. (2017) has serious problems, at least for the Amazon basin. In general, remote sensing of peatlands is difficult.

Peatlands have been the focus of numerous studies in recent years (Gumbricht et al., 2017; Xu et al., 2018; Melton et al., 2022; Global Peatland Map 2.0, 2022; Minasny et al., 2024), with multiple questions about the accuracy and evaluation of existing datasets. We added a paragraph in the section 5.2.3 Peatland integration to discuss this (lines 478 to 487 of the track changes document) :

*"Non-inundated peatlands account for a significant proportion of the GIEMS-MC$_{ISW+P}$ in terms of maximum area (50.2% of the MAmax, see Table 2). The peatland map used in this study to derive GIEMS-MC$_{ISW+P}$ comes from GLWDv2, and is a composite product based on four different estimates (see section 3.1.7 or Lehner et al. (2024a)). It should be emphasized that although this product is based on the best current knowledge, there is still limited consensus in the literature on the extent of peatlands. Mapping of peatlands remains challenging, as it is primarily a soil characteristic related to organic carbon content that cannot be directly detected by satellite data. Consequently, mapping efforts rely on approaches based on observations (in situ data and / or remote sensing proxies for, e.g., vegetation or hydrological data) to map their extent using models or machine learning. This task is particularly difficult in tropical regions, where in situ data are sparse and where dense cloud and vegetation covers limit remote sensing observations. This results in fewer regional maps over the Tropics with frequent revisions towards higher estimates of tropical peatland areas (Dargie et al., 2017; Hastie et al., 2024)."*

Though the ERA product provides a global estimate of the occurrence of snowcover, a comparison of these estimates with the SNODAS products for parts of North America or with other snowcover products would be useful.

Note: As Referee 2 asked a similar question, we have given the same answer to both reviewers.

The ERA5 snow mask is used in the production of GIEMS-2, and for consistency, we applied the same mask in GIEMS-MC. ERA5 offers the advantage of global coverage and an uninterrupted long-term record (from 1970 to the present, with ongoing updates). In our processing, ERA5 snow data is used to filter out pixels affected by snow, as snow has complex and highly variable behavior in passive microwave observations, which are used

to estimate surface water. The goal is to prevent any contamination of surface water estimates by snow.

Our approach is deliberately conservative in identifying snow-covered areas, which may result in missing some regions near the snow margin. Yet, this is expected to have minimal impact on global methane emission estimates, as temperatures in these areas are typically low (close to 0°C).

We acknowledge that ERA5 snow cover is not a perfect dataset. However, it has been found to be more consistent in terms of trends than, for instance, the NOAA CDR reanalysis product (Urraca et al., 2023). ERA5 effectively captures interannual variations (Kouki et al., 2023), and after 2004, it has been shown to provide the highest accuracy compared to ground measurements among available datasets (Urraca et al., 2023). Some discontinuities in the time series have been reported, particularly around 2004 (Urraca et al., 2023). However, we verified that this does not impact GIEMS-MC surface water extent in northern basins where the snow mask has the greatest influence. For example, no discontinuity is observed in 2004 in the Ob basin, as illustrated in Figure 9 of the paper. It is worth noting that ERA5 generally estimates a larger snow cover extent than other datasets, primarily due to higher values in mountainous regions (Kouki et al., 2023). However, these regions are also poorly represented in other long-term datasets, including remote sensing products (Bormann et al., 2018).

We added in section *2.4 Snow-covered pixel masking* of the manuscript a paragraph to discuss the snow mask importance (lines 498 to 503 of the track changes document):

*"For consistency with the snow mask used in GIEMS-2 production, we have used the same mask here in the GIEMS-MC generation. This mask is derived from the ERA5 product, which might overestimate the extent of snow cover but still captures interannual changes well (Kouki et al., 2023). The snow mask in GIEMS-MC is only a filter for pixels potentially contaminated by the presence of snow. The potential overestimation of snow cover extent should have limited implications for methane emissions, as methane emissions in these regions during the snow season should be a small fraction of global emissions, as discussed above."*

The European Space Agency Land Cover dataset has limitations when applied to seasonally varying wetland vegetation, and these uncertainties need to be mentioned.

Here, the ESA Land Cover dataset is used to estimate vegetation types for potential application in methane emission modeling (only the main vegetation type is provided, as an indicator, per 0.25°x0.25° pixel). Studies have shown that vegetation type highly influences methane emissions in wetland areas (Pangala et al., 2017; Vroom et al., 2022; Ge et al., 2024; Feron et al., 2024; Girkin et al., 2025). To support this, GIEMS-MC provides a vegetation and wetland type information, enabling potential refinement of methane emission parameterization based on vegetation characteristics. ERA5 CCI LC product has its limitations, but here only a yearly value at low resolution is given as indicator, the seasonal and inter-annual changes are not taken into account.

---

## Author Comment (AC2)

**Replies to Referee 2**

In this manuscript, the authors presented a new dataset of inundation dynamics, GIEMS-MethaneCentric. The authors improved the previous dataset by using updated input data and a revised systematic data production process. They compared the new data with other data (WAD2M based on SWAMPS) and obtained consistent results.

Major comments

Accurate inundation (methane-emitting aquatic surface) maps are undoubtedly important for the emission evaluation of methane, a potent greenhouse gas and short-lived climate forcer. However, uncertainties in the inundation dataset have been a serious problem in the global methane budget. The dataset presented in this study is remarkable, because it captured spatial heterogeneity by using updated satellite data and land surface maps. The GIEMS-2 data also covers a long period from 1992 to 2020, allowing us to assess interannual to decadal dynamics of inundation and resultant methane emissions. Moreover, in producing the dataset, the authors used new freshwater and paddy field data to avoid double counting, which is a serious problem in the global methane budget. The dataset is clearly useful for wetland and methane researchers; the spatial resolution (quarter degree) and time step (monthly) may be coarse for field studies but useful for regional to broader assessments.

Thank you for your comments and the time taken to review this manuscript. Hereafter, point-by-point replies to your concerns, and the corresponding changes in the manuscript.

I have a minor concern about the quality of the dataset. Namely, the results (e.g., Figure 3) show that peatlands made a substantial contribution to the global extent especially in tropical and northern latitudes. Nevertheless, the peatland extent was derived from static maps like PeatMap, and therefore interannual variability in peatland inundation could be underrepresented in the dataset. For example, in Figure 8, the anomalies were not largely different between GIEMS-MC$_{ISW}$ and GIESM-MC$_{ISW+P}$ except for variability due to snow cover in northern areas (Ob). In section 5.2.3., the authors discussed a problem with peatland integration but focused on the separation of inundated and saturated areas. As discussed in section 5.2.2. about rice paddy fields, the authors should discuss the temporal variability of peatlands; this can be serious in Southeast Asia (see Figure 7), where peatlands are prevailing and meteorological variability like ENSO is influential.

The peatland inundation variation should be captured in GIEMS-MC$_{ISW}$ as peatlands are not removed from inundated surfaces (inundated peatlands are included in inundated wetland definition). Indeed, inundated peatlands are detected by GIEMS-2. The difference between GIEMS-MC$_{ISW}$ and GIEMS-MC$_{ISW+P}$ is the integration of non inundated peatlands.

Overall, the manuscript is well prepared. The methodological description is adequate, and the dataset is presented nicely. I conclude that the manuscript is acceptable for publication after minor revision.

Minor comments

- I agree that the ERA5 is widely used meteorological dataset, but I am not sure the quality of snow density and depth in the dataset. Did you check it by comparing with observational data?

Note: As Referee 1 asked a similar question, we have given the same answer to both reviewers.

The ERA5 snow mask is used in the production of GIEMS-2, and for consistency, we applied the same mask in GIEMS-MC. ERA5 offers the advantage of global coverage and an uninterrupted long-term record (from 1970 to the present, with ongoing updates). In our processing, ERA5 snow data is used to filter out pixels affected by snow, as snow has complex and highly variable behavior in passive microwave observations, which are used to estimate surface water. The goal is to prevent any contamination of surface water estimates by snow.

Our approach is deliberately conservative in identifying snow-covered areas, which may result in missing some regions near the snow margin. Yet, this is expected to have minimal impact on global methane emission estimates, as temperatures in these areas are typically low (close to 0°C).

We acknowledge that ERA5 snow cover is not a perfect dataset. However, it has been found to be more consistent in terms of trends than, for instance, the NOAA CDR reanalysis product (Urraca et al., 2023). ERA5 effectively captures interannual variations (Kouki et al., 2023), and after 2004, it has been shown to provide the highest accuracy compared to ground measurements among available datasets (Urraca et al., 2023). Some discontinuities in the time series have been reported, particularly around 2004 (Urraca et al., 2023). However, we verified that this does not impact GIEMS-MC surface water extent in northern basins where the snow mask has the greatest influence. For example, no discontinuity is observed in 2004 in the Ob basin, as illustrated in Figure 9 of the paper. It is worth noting that ERA5 generally estimates a larger snow cover extent than other datasets, primarily due to higher values in mountainous regions (Kouki et al., 2023). However, these regions are also poorly represented in other long-term datasets, including remote sensing products (Bormann et al., 2018).

We added in section *2.4 Snow-covered pixel masking* of the manuscript a paragraph to discuss the snow mask importance (lines 498 to 503 of the track changes document):

*"For consistency with the snow mask used in GIEMS-2 production, we have used the same mask here in the GIEMS-MC generation. This mask is derived from the ERA5 product, which might overestimate the extent of snow cover but still captures interannual changes well (Kouki et al., 2023). The snow mask in GIEMS-MC is only a filter for pixels potentially contaminated by the presence of snow. The potential overestimation of snow*

*cover extent should have limited implications for methane missions, as methane emissions in these regions during the snow season should be a small fraction of global emissions, as discussed in the previous section."*

- Line 223: I agree to apply a clearing process to reduce the artifacts in coastal areas. However, I suspect that it resulted in the removal of riverine estuaries where are potentially important methane sources. Is my understanding correct?

Your understanding is correct, estuaries, deltas (such as lakes, rivers) are methane emitting areas that are not considered in GIEMS-MC's two dynamic wetland maps (ISW and ISW+P). In fact, estuaries and deltas are coastal areas and then suffer from the limitation of microwave observations in the coastal area (ocean contamination). To assess estuarine methane emissions in particular, and coastal methane emissions in general, users should make better use of other products with higher resolution. For example, optical products derived from MODIS could provide dynamic observations over coastal areas and sparse vegetation. GLWDv2 also provides static information on estuaries and deltas. In response to two reviewers comments, estimates of estuaries and deltas from GLWDv2 are added to the GIEMS-MC variables, following the same approach used for lakes, rivers, and reservoirs, (GIEMS-MCv1.1).

Note : We realized that only estuaries and not deltas had been removed as open surface waters in the GIEMS-MC process. This has been changed in the new manuscript and database. This modification is minor (~2% and ~1% in terms of MAmax on GIEMS-MC$_{ISW}$ and GIEMS-MC$_{ISW+P}$) and does not change the conclusions.

- Figure 3: Please show the period for the data used in the figure.

This has been added.

- Line 484: Can you give a rough estimation of how much the new dataset improves the evaluation of global wetland methane emissions? For example, a +10% larger inundation area may lead to correspondingly larger emissions.

GIEMS-MC provides approximately the same mean global wetland extent as WAD2M : the absolute global wetland methane emission may then depend mainly on the methane emission model used and its scaling factor. GIEMS-MC should hopefully be used in the Global Methane Budget to assess the impact of the new map on CH4 emission estimates. Compared to existing datasets, the key advantage of GIEMS-MC lies in its improved temporal variations, capturing seasonal and interannual dynamics without discontinuity, while also enhancing spatial patterns. These improvements can enhance methane budget estimates, particularly if new gridded wetland emission datasets based on GIEMS-MC are integrated into inversion models.  We have revised the paragraph to better highlight the relevance of GIEMS-MC for the methane research community (lines 530 to 533 of the track changes document):

*"Despite significant advances in methane measurement and modelling, accurate mapping of wetland extent remains a key challenge. In this context, we introduce GIEMS-MethaneCentric (GIEMS-MC). It is a new product that improves the temporal variability of the wetland extent by accurately capturing seasonal and interannual dynamics without discontinuities, while also enhancing spatial patterns."*

*"Despite significant advances in methane measurement and modelling, accurate mapping of wetland extent remains a key challenge. In this context, we introduce GIEMS-MethaneCentric (GIEMS-MC). It is a new product that improves the temporal variability of the wetland extent by accurately capturing seasonal and interannual dynamics without discontinuities, while also enhancing spatial patterns."*

---

## Author Comment (AC3)

**Replies to Referee 3**

This study presents an updated global inundation area dataset spanning 1992 to 2020, providing monthly data at a spatial resolution of 0.25°×0.25°. The dataset differentiates coastal areas and rice paddies, thereby mitigating the overestimation of wetland areas and improving data accuracy. This product has significant value for quantifying methane emissions from global wetlands. Overall, the article is concise and well-structured, accordingly I have only several comments requiring further clarification:

We are grateful for your review of the manuscript and for your comments, which we have responded to below.

Both peatland area and methane emission intensity are highly influenced by soil moisture saturation and water table. Thus, water table playing a crucial role in peatland area variation. How does the study address this issue?

We are aware that the water table is an important parameter for peatland/wetland methane emissions. The literature on wetland methane emission modelling presents two main approaches to account for wetland extent (Bohn and Lettenmaier, 2010):

- Method 1 (*uniform scheme*): Use of a static wetland extent (maximum wetland extent). In this case, the methane emission of the wetland is determined by the water table over the wetland parts of the whole pixel (> 0.5°x0.5°). e.g., Walter et al., 2001.
- Method 2 (*wet-dry scheme*): Use of a dynamic wetland extent, which constrains the wetland area that is inundated or saturated, excluding "dry" zones. e.g., UpCH4 (McNicol et at, 2023).

Both methods are simplifications with their own advantages and disadvantages. Method 1 must use modelled groundwater levels, as this variable cannot be observed globally. In addition, this method neglects the effects of spatial heterogeneity in water table depth (and therefore methane emissions), as considering water table at the resolution of land surface models (>25 km) is less relevant for wetland hydrology (Bohn and Lettenmaier, 2010).

Method 2 can use remote sensing based products such as GIEMS-2/GIEMS-MC. However, it neglects emissions from "drier" unsaturated topsoil wetlands, which depend on the depth of the water table.

The use of GIEMS-MC would be useful in Method 2, with its advantages and disadvantages. GIEMS-MC detects inundated but also soil surface saturation, but it does not provide direct information about water table depth in unsaturated areas. A possible useful information to enhance these two approaches would be a high resolution global dynamic product of the water table to produce a per pixel distribution of the water table in wetlands. A high resolution product of water table depth was proposed by Fan et al. (2013) for 2004 (the only global one to our knowledge), but 1. it was only seasonal (for 1 year)

and 2. it showed inconsistent patterns with GIEMS and other surface water datasets. At this stage, we decided not to include it.

GIEMS-MC users could use additional information to model methane emissions from other sources, such as embedded groundwater level distribution from a land surface model or future high-resolution groundwater products. GIEMS-MC provides an observation that could be used by land surface models to improve their internal simulations of groundwater depths, improving on the approach from Fan et al. (2013).

As shown in Figure 3, peatlands contribute significantly to wetland areas in high-latitude and tropical regions. In high-latitude areas, snow cover changes can partially explain the interannual variability of peatlands, but this approach is clearly unsuitable for low-latitude regions. How can the variability in low-latitude peatlands be better understood? Could this issue be discussed further?

Regarding the variations of peatland inundation, these variations should already be included in GIEMS-MC$_{ISW}$ (Inundated and Saturated Wetlands): the variations in the inundated peatland are detected by GIEMS-2 (as all other water surfaces) and including in the GIEMS-MC$_{ISW}$ dataset. Only in GIEMS-MC$_{ISW+P}$, all peatlands are included, inundated or not. The seasonal variability of GIEMS-MC$_{ISW+P}$ visible in Figure 3.a (color filling) is consistently smaller than that of GIEMS-MC$_{ISW}$, as GIEMS-MC$_{ISW+P}$ should theoretically not account for changes in peatland inundation (all peatlands are included).

Would subtracting coastal area directly from wetland areas result in the underestimate of wetland area by excluding riverine estuaries?

The passive microwave satellite data used to derive the surface water extent are very sensitive to all water surfaces, including ocean surfaces. Over coastal areas, the effect of the ocean surface can extend further inland, due to the detection of ocean contamination by the antenna side lobes. To minimize ocean contamination, a conservative solution is suggested by firstly suppressing coastal regions with a high ocean fraction (>10%) and secondly correcting of the coastal pixels. The reviewer is right that this process excludes estuaries and deltas, which are considered separately from wetlands in GIEMS-MC. In response to two reviewers comments, the static estimates of estuaries and deltas from GLWDv2 are now added to the GIEMS-MC variables, following the same approach used for lakes, rivers, and reservoirs, (GIEMS-MCv1.1).

Rainfed rice exhibits seasonal and interannual variations in their inundation status, though its contribution to total wetland area is relatively minor. Could this be discussed further?

This was added to the discussion about rice uncertainties Section 5.2.2 (lines 437 to 446 of the track changes document) :

*"Finally, subtracting the MIRCA2000 climatology in the GIEMS-MC processing and not taking into account the inter-annual variation of rice paddies and their inter-annual changes in inundation over the period*

*1992-2020 can lead to misclassification of rice paddies as wetlands (and the opposite). First, rainfed rice paddies in particular show significant inter-annual changes in inundation. Furthermore, the surface covered by rice paddies also changes over the years. The MIRCA2000 product is compared in Fig. 11 with the estimates from the Food and Agriculture Organization (FAO) of the United Nations estimates FAOSTAT (https://www.fao.org/faostat/en/#data/QCL, access 30/06/2023). FAOSTAT is widely used for global estimates of methane emissions from rice paddies, notably in the Emissions Database for Global Atmospheric Research (EDGAR ; Janssens-Maenhout et al. (2019)). The cropland area of rice paddies is increasing in South-East Asia, with FAOSTAT estimating +60 103 km2 between 1992 and 2020 in this region, which corresponds to the increasing trend of ~+50 103 km2 in GIEMS-MC$_{ISW}$ over this period (Sect. 4.3.2)."*

Finally, could the article delve into the specific advantages of using an update dataset that classifies inundation areas into distinct types for accurately estimating global wetland area and its methane emissions?

We believe that the point about the benefit of having harmonized maps to avoid problems of double counting and non-counting between wetlands and other inundated areas has been mentioned throughout the text (Introduction lines 35-38 of the track changes document, Rice discussion lines 450-454 and Conclusion lines 545-546).

References have been added in section 3.3 to justify the need of vegetation/wetland types consideration (lines 280 to 281 of the track changes document) :

*"This vegetation information is added in GIEMS-MC as it is expected to help improve the estimation of wetland methane emissions (Pangala et al., 2017; Vroom et al., 2022; Feron et al., 2024; Girkin et al., 2025; Ge et al., 2024)."*

We have also changed a sentence in the conclusion about the interest of also distinguishing wetland and vegetation types by adding these variables in GIEMS-MC (lines 543 to 544 of the track changes document):

*"Information on the dominant vegetation and wetland type for each pixel is also provided, as these factors help improve the understanding and accurate modeling of methane emissions from wetlands."*